

# Enhancing Volcanic Eruption Simulations with the WRF-Chem v4.8

Alexander Ukhov[1], Georgiy Stenchikov[1], Jordan Schnell[2,3], Ravan Ahmadov[2], Umberto Rizza[4],
Georg Grell[2], and Ibrahim Hoteit[1]

[1]King Abdullah University of Science and Technology, Division of Physical Sciences and Engineering, Thuwal, Saudi Arabia
[2]NOAA Earth System Research Laboratory, Boulder, CO, USA
[3]CIRES, University of Colorado, Boulder, CO, USA
[4]National Research Council—Institute of Atmospheric Sciences and Climate (CNR-ISAC), 73100 Lecce, Italy

**Correspondence:** Ibrahim Hoteit (ibrahim.hoteit@kaust.edu.sa)

**Abstract.**

Volcanic eruptions are one of the major natural hazards, exerting profound effects on the environment and climate. The emissions associated with such eruptions pose substantial risks to terrestrial systems and public health, particularly through the induction of acid rain and air pollution. Volcanic ash influences populations at distances reaching several thousand kilometers from the erupted volcano. Also, accurate forecasting of volcanic clouds is crucial for air traffic control. This study introduces enhancements in the simulation of volcanic eruptions and the transport of volcanic material using the Weather Research and Forecasting model coupled with Chemistry (WRF-Chem) v4.8 model. Improvements include the addition of wet and dry deposition of ash and sulfate, improved $SO_2$ chemical transformation mechanisms, and corrections of the gravitational deposition of ash. Ash, sulfate, and $SO_2$ mass budget analyses were conducted. Furthermore, we included the direct radiative effect of ash and sulfate aerosols. Additionally, we developed an open-source Python-written emission preprocessor called PrepEmisSources to facilitate and streamline the preparation of volcanic emissions. Accordingly, the model code was extended to simulate complex volcanic emission scenarios using the emissions file prepared using the PrepEmisSources tool. The results suggest that the enhanced WRF-Chem v4.8 code provides an accurate representation of volcanic ash, $SO_2$, and sulfate dispersion, deposition, and chemical transformation. These improvements will aid in volcanic debris forecasting and will allow for the use of the model for assessments of volcanic aerosols on climate and for geoengineering problems, including modeling of stratospheric aerosol injection.

## 1 Introduction

Volcanic eruptions are one of the major natural hazards. The associated volcanic emissions pose substantial risks to agriculture, infrastructure, and public health, particularly through the tephra fallout, the induction of acid rain, and air pollution. Additionally, these emissions impact the climate by releasing sulfur dioxide ($SO_2$), which subsequently undergoes conversion into sulfate aerosols due to oxidation by hydroxyl radicals (OH) and hydrogen peroxide ($H_2O_2$). Volcanic aerosols influence extensive populations at distances reaching several thousand kilometers from the erupted volcano (Stewart et al., 2022). More-





over, information on ash concentration and the location of the volcanic ash cloud is essential for air traffic control (Casadevall, 1994). Considering these aspects, accurate numerical modeling of the transport and deposition of volcanic debris is essential.

Volcanic ash transport and dispersion models (VATDs) are widely used to forecast the dispersion and transport of ash clouds over hours to days to define hazards to aircraft and to communities downwind. These models are being used by nine Volcanic Ash Advisory Centers (VAACs) worldwide. VAACs provide forecasts on the expected presence of volcanic ash in the atmosphere to mitigate the risk to aviation. There are several VATD models; those include Euler and Lagrangian types of models, such as PUFF (Searcy et al., 1998), HYSPLIT (Stein et al., 2015), NAME (Jones et al., 2007), FLEXPART (Stohl et al.,

2005), FALL3D (Folch et al., 2009). These models are also useful in forecasting areas to be impacted by tephra fall. Most of the VATD models are 'offline,' since they separately describe the physics and chemistry characterizing the dispersion of volcanic emissions in the atmosphere. Although 'offline' models are fast, they do require the meteorological fields, which need to be computed in advance. These models do not fully capture interactions between meteorology and aerosol dynamics. Specifically, the interplay between volcanic aerosols and solar as well as terrestrial radiation exemplifies this form of interaction, which

is not accounted for in the 'offline' models. For example, important at the initial stage of the eruption, ash particles have a relatively short lifetime. Erupted into the atmosphere, ash inhibits sunlight from reaching the Earth's surface, cooling the surface and warming the ash cloud due to strong absorption. In contrast, water vapor and sulfate are important in further stages of volcanic cloud dispersion. The stratospheric sulfate aerosol clouds can persist from a few months to a couple of years, reflecting solar radiation into space, inducing Earth's surface cooling (Stenchikov et al., 1998). The absorption of infrared

upwelling radiation warms the layer where sulfate particles reside. In turn, being the most abundant greenhouse gas, water vapor plays a crucial role in global warming. Like $CO_2$ and $CH_4$, it absorbs and re-emits infrared radiation, trapping heat in the atmosphere and warming the planet. Therefore, a fully coupled 'online' meteorology–chemistry model is a necessary tool to account for radiative feedback in simulating volcanic aerosols and meteorological fields.

## 1.1   Previous work

Among the available "online" tools, the coupled Weather Research and Forecasting model coupled with Chemistry (WRF-Chem) (Skamarock et al., 2005; Grell et al., 2005) open-source Eulerian model is distinguished for its capability to simulate atmospheric chemistry, air quality, transport, and deposition of aerosols, including volcanic ash. Webley et al. (2012), Stuefer et al. (2013), and Steensen et al. (2013) were the first who included in the WRF-Chem model the capability to simulate emissions from volcanoes and to predict the transport and concentration of ash and $SO_2$. In the model, ash is gravitationally

settled, and $SO_2$ undergoes oxidation to sulfate using the prescribed OH vertical distribution. Stuefer et al. (2013) introduced ash size distribution classification and extended the volcanic emission preprocessor with time-variant emissions, which can either be specified directly as mass fluxes or calculated with the formula that relates plume height to the mass emission rate (Mastin et al., 2009). Webley et al. (2012) provided a detailed evaluation of the eruption of Eyjafjallajökull in Iceland in 2010. Steensen et al. (2013) performed a qualitative comparison of Mount Redoubt 2009 volcanic clouds using the PUFF and WRF-

Chem dispersion models and satellite remote sensing data. Hirtl et al. (2019) simulated the 2010 Eyjafjallajökull volcanic eruption to study ash dispersion and to highlight the importance of the radiative feedback effect on the meteorological fields. In



particular, with enabled radiative feedback, a better agreement of volcanic cloud location with radiosonde measurements was achieved. Hirtl et al. (2020) made further extensions of the emission preprocessor introduced by Stuefer et al. (2013) to integrate complex sources. In particular, the volcanic preprocessor was integrated into the WRF-Chem code to allow the temporally and

vertically resolved input data for the simulation of the eruption of the Grimsvötn volcano in Iceland in May 2011. Unfortunately, this code is not a part of the official release. Rizza et al. (2023) simulated a sequence of Mt Etna paroxysms, coupling the WRF-Chem with near-source L-Band radar observations. Egan et al. (2020) analyzed tephra fallout and in situ airborne measurements of ash from the eruption of Eyjafjallajökull volcano in 2010. de Bem et al. (2024) used WRF-Chem to simulate ash transport from the eruption of Chile's Calbuco volcano that happened in 2015. In Stenchikov et al. (2021), a modified version of the

WRF-Chem model was used to evaluate the radiative effects of volcanic ash, sulfate, $SO_2$, and water vapor emitted during the Mt. Pinatubo eruption. They used the Goddard Chemistry Aerosol Radiation and Transport (GOCART) aerosol module (Chin et al., 2002) to represent ash as mineral dust with a modified complex refractive index (RI). The same approach was used in Ukhov et al. (2023) for inverse modeling of emission rates of ash and $SO_2$. Stenchikov et al. (2025) investigated the recent eruption of the Hunga Tonga-Hunga Haapai underwater volcano on 15 January 2022, which released 150 Mt of water vapor

into the stratosphere. Water vapor strongly affected the dynamics of the aerosols as a result of the radiative cooling of the water vapor cloud. In their work, a two-moment aerosol microphysical module named Model for Simulating Aerosol Interactions and Chemistry (MOSAIC) (Zaveri et al., 2008) coupled with the Carbon Bond Mechanism Version Z (CBM-Z) chemistry module (Zaveri and Peters, 1999) was used.

Although the WRF–Chem model has been extensively utilized over the past decade and has demonstrated significant efficacy

as a tool for forecasting the transport and dispersion of volcanic ash and $SO_2$ (Ukhov et al., 2020b, 2025), it still lacks a set of parameterizations. Furthermore, the interaction of ash and sulfate with solar and terrestrial radiation for the simulation of the direct radiative effect has not been implemented explicitly into the WRF-Chem's code so far. There were only implicit attempts to account for only ash radiation interaction, ignoring the sulfate. In particular, Hirtl et al. (2019) redistributed the three finest ash bins into PM2.5 3D field, which was used in the calculation of the optical properties. This methodology presupposes a

constant size distribution of PM2.5, which constitutes a rather crude assumption considering the variability in the ash size distribution within the cloud, particularly in the initial days. In addition, the standard methodology for preparing volcanic emissions files using the open-source tool PREP-CHEM-SRC (Freitas et al., 2011) is notably cumbersome and not flexible.

Here, we rectify the existing imperfections and add new capabilities to the WRF-Chem code v4.8 related to the simulation of volcanic eruptions. In particular, we account for major sinks: wet and dry deposition of ash, $SO_2$, sulfate, and $SO_2$ oxidation

by OH and $H_2O_2$. We found and fixed and error in the subroutine for ash gravitational settling. Moreover, we implemented the capability to simulate direct radiative effects of ash and sulfate aerosols, acknowledging the substantial radiative forcing exerted by volcanic eruptions on the climate system (Stenchikov et al., 2021). Recent studies (Stenchikov et al., 2025) highlight the importance of water vapor in modulating the dynamics of the volcanic plume. Therefore, simulation of emissions of water vapor and sulfate aerosol was also added. In addition, we developed an open-source emission preprocessor called PrepEmis-

Sources (Ukhov and Hoteit, 2025). In contrast to the standard PREP-CHEM-SRC utility, our tool enhances the workflow and



provides increased flexibility in defining the volcanic eruption process based on the eruption source parameters. More details are presented in Appendix A.

## 2 Code modifications

Most of the added code was adapted from the GOCART aerosol module (Chin et al., 2002) implemented in WRF-Chem. The
GOCART module simulates major tropospheric aerosol components, including sulfate, dust, black and organic carbon, and sea-salt, and includes algorithms for dust and sea salt emissions, dry deposition, and gravitational settling and oxidation of $SO_2$ (Ukhov et al., 2020a). Below, we describe the changes that we implemented in the WRF-Chem v4.8. Certain subsections incorporate pseudocode designed to compute some output diagnostics enumerated in Table B1. This inclusion is attributable to the absence of the necessary calculations for these diagnostics within the corresponding subroutines in the GOCART aerosol
module.

### 2.1 SO$_2$ oxidation by OH and H$_2$O$_2$

We consider the chemical depletion of $SO_2$ resulting from oxidation processes. We replicate the code from the GOCART module, with modifications in the parameters used for the oxidation reaction with OH. The oxidation by $H_2O_2$, which had not been previously utilized in modeling volcanic $SO_2$, has been implemented within our study. See subroutine *go-*
*cart_volc_chem_driver()* in *module_volc_chem.F* for details. The oxidation is activated by setting *gaschem_onoff*=1 in the *namelist.input* file. The OH and $H_2O_2$ fields are not computed interactively; instead, they are prescribed based on zonal and monthly mean OH and $H_2O_2$ fields (Liu et al., 2023) derived from the simulation of the Chemical Lagrangian Model of the Stratosphere (CLaMS) (Pommrich et al., 2014) and Copernicus Atmosphere Monitoring Service (CAMS) reanalysis (Inness et al., 2019), respectively. The Supplementary Material includes two Python scripts for interpolating OH and $H_2O_2$ fields into
the WRF-Chem domain.

Three-body second-order reaction rate coefficient ($k$) of the $SO_2$-OH oxidation process: OH + $SO_2$ + M → sulfates + ... is calculated according to Table 2-1 in Burkholder et al. (2020) as follows:

$$k = \frac{k_0 \cdot M}{1 + \frac{k_0 \cdot M}{k_\infty}} \times 0.6^{\left(1/\left(1+\log_{10}\left(\frac{k_0 \cdot M}{k_\infty}\right)^2\right)\right)} \text{ (cm}^3 \text{ molecule}^{-1} \text{ s}^{-1}), \tag{1}$$

where $k_0 = 2.9 \cdot 10^{-31} \times \left(\frac{298}{T}\right)^{4.1}$ (cm$^6$ molecule$^{-2}$ s$^{-1}$) is the low-pressure limit rate,


$k_\infty = 1.7 \cdot 10^{-12} \times \left(\frac{298}{T}\right)^{-0.2}$ (cm$^3$ molecule$^{-1}$ s$^{-1}$) is the high-pressure limit rate,

$T$ is the temperature (K), and $M$ is the air density (molecules cm$^{-3}$).



The updated $SO_2$ concentration $(mol\ mol^{-1})$ is calculated as follows:

$$SO_2 = SO_2 \times \exp\left(-k \cdot OH \cdot \Delta t\right), \tag{2}$$

where OH concentration is given in $(molecules\ cm^{-3})$, and $\Delta t$ is a time step (s). In general, the production of OH is driven by the photolysis of ozone, which causes a pronounced correlation with the diurnal cycle. To address this variability, the concentration of OH is multiplied by a scaling factor, which depends on the solar zenith angle.

In-cloud oxidation by hydrogen peroxide $(H_2O_2)$ is another chemical sink for $SO_2$. Here we replicate the code implemented in the GOCART module. The loss of $SO_2$ due to aqueous-phase oxidation by $H_2O_2$ in clouds is parameterized following a simple cloud-fraction-weighted approach. The scheme applies only for grid cells where the temperature exceeds 258 K and the cloud fraction $(f_c)$ is nonzero. To avoid excessive depletion when $SO_2$ exceeds the available $H_2O_2$, the effective cloud fraction $f_c^*$ is scaled:

$$f_c^* = \begin{cases} f_c \times \frac{H_2O_2}{SO_2}, & \text{if } SO_2 > H_2O_2. \\ f_c, & \text{otherwise} \end{cases} \tag{3}$$

$SO_2$ and $H_2O_2$ are given in $(mol\ mol^{-1})$. The post-reaction $SO_2$ is:

$$SO_2 = SO_2 \times (1 - f_c^*). \tag{4}$$

This simple scheme ensures that aqueous $SO_2$ oxidation is limited by both cloud coverage and oxidant availability, without explicit kinetics of dissolution and reaction in cloud water.

## 2.2 Convective scavenging

The process of ash and sulfate scavenging through convective precipitation is incorporated within subroutine *grelldrvct()* implemented in *module_ctrans_grell.F*. This option is activated by setting *conv_tr_wetscav*=1. The convective scavenging for $SO_2$ is not calculated, taking into account the assumption that all $SO_2$ undergoes oxidation within clouds. This process was not accounted for before in the simulation of ash and sulfate.

## 2.3 Large scale scavenging

The code for large scale scavenging of ash, sulfate and $SO_2$ is based on the one used for GOCART aerosols and gases. We implemented the code in the subroutine *wetdep_ls_volc()* in the *module_vash_settling.F* file. This option is activated by setting *wetscav_onoff* to any negative number. The scavenging process is applied to ash, $SO_2$ and sulfate. This scheme has a tuning parameter $\alpha$, which is the scavenging efficiency. We set $\alpha$=0.5 for ash and $\alpha$=1 for $SO_2$ and sulfate, due to their high solubility and efficient removal in cloud water. We added the corresponding diagnostics, which reflect the accumulated amount of scavenged ash, $SO_2$ and sulfate:

$$WD\_ASH\_SC = WD\_ASH\_SC + \sum_{i=1}^{10} [dvash\_i] \cdot \rho \cdot \Delta z \cdot 10^{-6} (g\ m^{-2}), \tag{5}$$





$$WD\_SO2\_SC = WD\_SO2\_SC + [dSO_2] \cdot \rho \cdot \Delta z/28.97 \ (\text{mmol m}^{-2}), \tag{6}$$

$$WD\_SULF\_SC = WD\_SULF\_SC + [dsulf] \cdot \rho \cdot \Delta z/28.97 \ (\text{mmol m}^{-2}). \tag{7}$$

These formulas are applied for each layer of the atmospheric column, where $\Delta z$ is the layer width, $[dSO_2]$, $[dsulf]$, $[dvash\_i]$ are the computed fractions of concentrations of $SO_2$ (ppmv), sulfate (ppmv), and ash (µg kg$^{-1}$) in the $i$th bin, respectively, subjected to scavenging within the timestep at a specific layer of the atmospheric column. $\rho$ is the dry air density (kg m$^{-3}$), 28.97 (g mol$^{-1}$) is the air molar mass. These diagnostic formulas were previously absent in the GOCART module; nonetheless, they can be integrated by adhering to our proposed methodology.

## 2.4 Dry deposition

Dry deposition of $SO_2$, sulfate, and ash was implemented by analogy with the GOCART module, where dry deposition is combined with vertical mixing and activated by setting $vertmix\_onoff$=1. We implemented the code in the *module_vash_settling.F* file in the *volc_ash_sulf_so2_drydep_driver()* subroutine. Dry deposition velocity is calculated according to Wesely (2007) and used as a boundary condition near the surface for the flux of the species. The dry deposition removes aerosols from the lowermost model layer as a dry deposition flux. The accumulated amounts of $SO_2$, sulfate, and ash are computed using the following expressions:

$$SO2\_DRYDEP = SO2\_DRYDEP + [SO_2] \cdot V_{dry,SO_2} \cdot \rho \cdot \Delta t \cdot 10^{-6}/ 28.97 \ (\text{mol m}^{-2}), \tag{8}$$

$$SULF\_DRYDEP = SULF\_DRYDEP + [sulf] \cdot V_{dry,sulf} \cdot \rho \cdot \Delta t \cdot 10^{-6}/ 28.97 \ (\text{mol m}^{-2}), \tag{9}$$

$$ASH\_DRYDEP = ASH\_DRYDEP + \left( \sum_{i=1}^{10} [vash\_i] \cdot V_{dry,vash\_i} \right) \cdot \rho \cdot \Delta t \cdot 10^{-9} (\text{kg m}^{-2}). \tag{10}$$

$[SO_2]$, $[sulf]$, $[vash\_i]$ are the corresponding concentrations of $SO_2$ (ppmv), sulfate (ppmv), and ash (µg kg$^{-1}$) in the $i$th bin, $\Delta t$ is the timestep (s), and $V_{dry,SO_2}$, $V_{dry,Sulf}$, and $V_{dry,vash\_i}$ are the computed dry deposition velocities (m s$^{-1}$) for ash particles in $i$th bin.

## 2.5 Gravitational settling of sulfate aerosols

As in the GOCART aerosol module, a bulk approach is used to represent sulfate aerosols. The sulfate aerosol droplets in the troposphere are smaller than 0.2 µm and therefore, in the troposphere, usually only dry and wet scavenging of sulfate aerosols is





calculated. In contrast, in the stratosphere, the sulfate aerosol droplets are bigger, air density is lower, and gravitational settling

becomes the leading deposition process. Therefore, we implemented the gravitational settling of sulfate aerosols following the

same assumptions as in Stenchikov et al. (2021). Sulfate gravitational settling numerical scheme is implemented and adapted

from the dust gravitational deposition scheme used in the GOCART scheme (Ukhov et al., 2021). Sulfate aerosol density is

$1800 \, \mathrm{kg \, m^{-3}}$. Based on observations Borrmann et al. (1995) and Dessler et al. (2014), we assume that sulfate aerosol number-

density size distribution can be approximated by Aitken and accumulation lognormal modes. Aitken mode has a median radius

$r_1 = 0.09 \, \mathrm{\mu m}$ and the geometric width $\sigma_1 = 1.4$. The accumulation mode has a median radius $r_2 = 0.32 \, \mathrm{\mu m}$ and the geometric

width $\sigma_2 = 1.6$. Only sulfate particles in accumulation mode are gravitationally settled, assuming their wet mean radius of

volume size distribution. Firstly, we determine dry radii $R_{\mathrm{dry,sulf}}$ calculated for sulfate aerosol volume median radii:

$$R_{\mathrm{dry,sulf}} = r_2 \cdot \exp\left(3 \cdot \log(\sigma_2)^2\right). \tag{11}$$

We obtain that $R_{\mathrm{dry,sulf}} = 0.62 \, \mathrm{\mu m}$ at given parameters of accumulation lognormal mode. Secondly, sulfate aerosol wet radius

($R_{\mathrm{wet,sulf}}$) is defined by computed $R_{\mathrm{dry,sulf}}$, relative humidity $RH$, and hygroscopic parameter $\kappa$:

$$R_{\mathrm{wet,sulf}} = R_{\mathrm{eff,sulf}} \cdot \left(\frac{1 + RH \cdot (\kappa - 1)}{1 - RH}\right)^{\frac{1}{3}}. \tag{12}$$

Following Petters and Kreidenweis (2007) and Aquila et al. (2012), we assume that $\kappa = 1.19$ for sulfate particles. This function-

ality is implemented in *module_vash_settling.F* subroutine *sulf_settling_driver()*. The accumulated amount of gravitationally

settled sulfate is computed as follows:

$$\mathrm{SULF\_GRAV\_SETL} = \mathrm{SULF\_GRAV\_SETL} + [\mathrm{SULF}] \cdot V_{settling,Sulf} \cdot \rho \cdot \Delta t \cdot 10^{-9} (\mathrm{kg \, m^{-2}}), \tag{13}$$

where [SULF] sulfate concentration ($\mathrm{\mu g \, kg^{-1}}$), and $V_{settling,Sulf}$ is the computed settling velocity ($\mathrm{m \, s^{-1}}$) of sulfate parti-

cles.

## 2.6 PM calculations

For convenience, we also added diagnostic output containing computed PM2.5 and PM10 surface concentrations. We modified

PM10 and PM2.5 calculations by analogy with Ukhov et al. (2021). The subroutine *sum_pm_gocart()* in *module_volc_chem.F*

calculates $PM_{2.5}$ and $PM_{10}$ surface concentrations using the following formulas:

$$PM2.5 = \rho \cdot (1.375 \cdot sulf + 0.672 \cdot vash\_10), \tag{14}$$

$$PM10 = \rho \cdot (1.375 \cdot sulf + vash\_10 + vash\_9 + 0.356 \cdot vash\_8), \tag{15}$$

Factor 1.375 compensates for missing $NH_4$, which neutralizes sulfate and produces ammonium sulfate. $sulf$ is sulfate

concentration converted from ppmv to $\mathrm{\mu g \, kg^{-1}}$, $\rho$ is the dry air density ($\mathrm{kg \, m^{-3}}$), $vash\_8, 9, 10$ are the mixing ratios ($\mathrm{\mu g \, kg^{-1}}$)

of the ash in the smallest three bins. Coefficients 0.672 and 0.356 are computed assuming that ash volume size distributions

are functions of the natural logarithm of the particle radius, see Ukhov et al. (2021) for details.





## 2.7 Ash gravitational settling

The gravitational settling of aerosols is a key driver of their removal from the atmosphere. The modeled volcanic ash is
subdivided into different bins representing the size spectrum of the particles. In WRF-Chem, ash particles are categorized into
10 size bins based on their radii, ranging from 0.01955 to 1000.0 μm, see Table 1. Ash can be modeled using 4 or 10 ash bins
(*chem_opt*=403 or *chem_opt*=400,402, respectively). For the ash gravitational settling, the terminal velocities are calculated
for the mean arithmetic radii ($R_m$) within each bin size. Ash density is assumed to be 2500 $\mathrm{kg\ m^{-3}}$, see Table 1.

### 2.7.1 Correction of terminal velocity for large ash particles

For particles with diameter D<10 μu the Stokes law (Stokes et al., 1851) along with the Cunningham slip correction factor $C_c$
(Cunningham, 1910) accurately describes the settling speed of particles. For such particles, the slip correction term dominates.
However, the settling speed of larger particles deviates substantially from Stokes' law. In WRF-Chem, particles' radii span
over several orders of magnitude. However, the required large particle drag correction is not accounted for in the code, which
leads to a strong overestimation of the settling velocities.

Therefore, based on the approach proposed in Mailler et al. (2023), we corrected the settling velocity for vash_1..6 bins
covering the mean arithmetic radii range from 500 to 23.44 μu. Their derivation is based on the Clift and Gauvin (1971)
drag coefficient formulation and provides an approximated expression of the large-particle drag correction factor. The $C_c$ is
estimated based on the Davies (1945) expression as follows:

$$C_c = 1 + Kn \left( 1.257 + 0.4 \exp\left( \frac{-1.1}{Kn} \right) \right), \tag{16}$$

where $Kn$ is the Knudsen number of the particle with diameter $D$:

| Bin | Radius lower bound (μm) | Radius upper bound (μm) | $R_m$ (μm) | Density (kg m$^{-3}$) |
|---|---|---|---|---|
| vash_1 | 500 | 1000 | 500 | 2500 |
| vash_2 | 250 | 500 | 375 | 2500 |
| vash_3 | 125 | 250 | 187.5 | 2500 |
| vash_4 | 62.5 | 125 | 93.75 | 2500 |
| vash_5 | 31.25 | 62.5 | 46.88 | 2500 |
| vash_6 | 15.625 | 31.25 | 23.44 | 2500 |
| vash_7 | 7.8125 | 15.625 | 11.72 | 2500 |
| vash_8 | 3.90625 | 7.8125 | 5.86 | 2500 |
| vash_9 | 1.95325 | 3.9065 | 2.93 | 2500 |
| vash_10 | 0.01955 | 1.95325 | 0.97 | 2500 |

**Table 1.** Ash particle bin size ranges with corresponding WRF-Chem ash bins.



$$Kn = \frac{2\lambda}{D}, \tag{17}$$

where $\lambda$ is the mean free path of molecules in air.

Stokes terminal velocity including the slip correction term is computed as follows:

$$\tilde{v}_\infty^{\text{Stokes}} = C_c \frac{D^2(\rho_p - \rho_a)g}{18\mu}, \tag{18}$$

where $\rho_p$ and $\rho_a$ are the particles' and air densities, respectively, and $g$ is the gravitational acceleration constant.

The following equation takes into account both the slip correction factor and the large-particle drag correction factor:

$$v_\infty = \tilde{v}_\infty^{\text{Stokes}} \times \left[ 1 - \left( 1 + \left( \frac{\tilde{R}}{2.440} \right)^{-0.4335} \right)^{-1.905} \right], \tag{19}$$

where parameter $\tilde{R}$ is defined as follows:

$$\tilde{R} = \frac{\rho_a D \tilde{v}_\infty^{\text{Stokes}}}{2\mu}, \tag{20}$$

For $\tilde{R} < 0.0116$, eq. 19 can be replaced as follows:

$$v_\infty = \tilde{v}_\infty^{\text{Stokes}}. \tag{21}$$

The accumulated amount of ash deposited by gravitational settling across all bins is determined as follows:

$$\text{ASH\_FALL} = \text{ASH\_FALL} + \left( \sum_{i=1}^{10} [vash\_i] \cdot V_{\infty, vash\_i} \right) \cdot \rho \cdot \Delta t \cdot 10^{-9} (\text{kg m}^{-2}), \tag{22}$$

where $[vash\_i]$ is the concentration of ash ($\mu$g kg$^{-1}$) in the $i$th bin, and $V_{\infty, vash\_i}$ are the computed settling velocities
(m s$^{-1}$) for ash particles in $i$th bin.



### 2.7.2 Corrections in ash settling

We found that in the WRF-Chem code, the gravitational settling of volcanic ash was calculated incorrectly. The finite-difference scheme (implemented in the subroutine *vsettling()* file *module_vash_settling.F*) does not account for the change in air density when the deposition mass flux is being calculated. Ash is usually emitted into high altitudes, where air density is less by two orders of magnitude with respect to the surface. Thus, in the course of the gravitational settling, the total ash in the atmosphere increases, violating the mass balances. We rectified this issue using a modified version of the finite-difference scheme, which conserves the mass of ash in the atmosphere. Below we describe the implemented changes.

The change of aerosol mass due to gravitational settling at downward directed settling velocity $w$ (m s$^{-1}$) is described using the conservative (flux) form for the integration of prognostic equations (Grell et al., 2005):

$$\frac{\partial(q\,m)}{\partial t} = \frac{\partial(q\,m\,w)}{\partial z}, \tag{23}$$

where $q$ is a ash mixing ratio (µg kg$^{-1}$), and $m$ is the dry air mass (kg). This equation can be discretized into the following form:

$$\frac{q_k^{n+1}\,m_k^{n+1} - q_k^n\,m_k^n}{\Delta t} = \frac{q_{k+1}^{n+1}\,m_{k+1}^{n+1}\,w_{k+1}^{n+1} - q_k^{n+1}\,m_k^{n+1}\,w_k^{n+1}}{\Delta z_k}, \tag{24}$$

where $\Delta z_k$ is the height of the $k$ model level, $\Delta t$ is the model time step. Subscript $k$ denotes the model levels and superscript $n$ is the time-level. Taking into account that the calculation of gravitational settling is split from the calculation of the continuity equation, we assume $m_k^{n+1} \approx m_k^n$ and get the following:

$$q_k^{n+1} - \frac{\Delta t}{\Delta z_{k+1}}\,q_{k+1}^{n+1}\,\frac{m_{k+1}^{n+1}}{m_k^n}\,w_{k+1}^{n+1} + \frac{\Delta t}{\Delta z_k}\,q_k^{n+1}\,w_k^{n+1} = q_k^n. \tag{25}$$

Rewriting eq. 25 taking into account that $m_k^n = \rho_k^n\,\Delta z_k \Delta x\,\Delta y$ and $m_{k+1}^{n+1} = \rho_{k+1}^{n+1}\,\Delta z_{k+1}\Delta x\,\Delta y$ (where $\rho$ and $\Delta x\,\Delta y$ are the dry air density (kg m$^{-3}$) and cell area (m$^2$), respectively) gives the following:

$$q_k^{n+1}\left(1 + \frac{\Delta t\ w_k^{n+1}}{\Delta z_k}\right) - \frac{\Delta t\,w_{k+1}^{n+1}}{\Delta z_{k+1}}\,q_{k+1}^{n+1}\left(\frac{\rho_{k+1}^{n+1}\,\Delta z_{k+1}\,\Delta x\,\Delta y}{\rho_k^n\,\Delta z_k \Delta x\,\Delta y}\right) = q_k^n. \tag{26}$$

Rearranging eq. 26 gives the following solution for $q_k^{n+1}$:

$$q_k^{n+1} = \left(q_k^n + \frac{\Delta t\,w_{k+1}}{\Delta z_k}\,q_{k+1}^{n+1}\frac{\rho_{k+1}}{\rho_k}\right)\left(1 + \frac{\Delta t\,w_k}{\Delta z_k}\right)^{-1}. \tag{27}$$

Eq. 27 is solved for each model column from the top to the bottom. For the topmost layer eq. (27) transforms into:

$$q_k^{n+1} = q_k^n\left(1 + \frac{\Delta t\,w_k}{\Delta z_k}\right)^{-1}. \tag{28}$$

Previous implementation of the finite difference scheme in the ash settling routine did not account for the dry air densities ratio in eq. (27). Consequently, when ash particles are settling from higher altitudes, a larger error is accumulated.





## 2.8 Inclusion of ash and sulfate into radiation calculation

The treatment of optically active ash and sulfate is vitally important. In particular, absorbing solar radiation, ash heats the atmosphere and is lofted (Stenchikov et al., 2021; Abdelkader et al., 2022). Ash has a relatively short lifetime and does 260 not affect climate much. While sulfate aerosols scatter solar radiation, have a longer lifetime, and play a primary role in aerosol–climate interactions.

By analogy with the subroutine *optical_prep_gocart()* being called for the calculation of volume-averaged refractive index (RI) when the GOCART aerosol module is being used, we implemented the subroutine *optical_prep_volc()* in *module_optical_averaging.F* file. This subroutine computes the volume-averaged RI needed for Mie calculations. The RI of ash = 265 $1.550 + i0.001$ in the shortwave spectral range has been chosen to approximate ash optical properties (Pollack et al., 1973; Carn and Krotkov, 2016). The larger imaginary part of the RI corresponds to the stronger absorption of solar radiation, enhancing atmospheric heating.

Vertical profiles of aerosol optical properties such as aerosol optical depth, single-scattering albedo, and asymmetry factor are computed by the parameterized Mie theory (Ghan and Zaveri, 2007) at 4 wavelengths (300, 400, 600, and 1000 nm). 270 Wavelength interpolation based on Angstrom coefficients for these 3 quantities is used as input for the Rapid Radiative Transfer Model (RRTMG) Iacono et al. (2008) for shortwave and longwave radiation options (*ra_lw_physics*=4 and *ra_sw_physics*=4). The Mie parameterization was modified by Fast et al. (2006) and Barnard et al. (2010) for the sectional representation of the aerosol size distribution, such that the Mie subroutine requires input of ash and sulfate concentration presented in eight intervals. These intervals are identical to those used in the MOSAIC microphysical module (Zaveri et al., 2008). Therefore, we 275 further refer to them as MOSAIC bins ($MOS_{1,2,3,4,5,6,7,8}$), see Table 2.

| Bin | Radius lower bound (μm) | Radius upper bound (μm) |
|---|---|---|
| $MOS_1$ | 0.01953125 | 0.0390625 |
| $MOS_2$ | 0.0390625 | 0.078125 |
| $MOS_3$ | 0.078125 | 0.15625 |
| $MOS_4$ | 0.15625 | 0.3125 |
| $MOS_5$ | 0.3125 | 0.625 |
| $MOS_6$ | 0.625 | 1.25 |
| $MOS_7$ | 1.25 | 2.5 |
| $MOS_8$ | 2.5 | 5.0 |

**Table 2.** Particle dry-radii range for the 8 size bins employed by MOSAIC.

As mentioned above, ash mass can be redistributed between 4 or 10 bins (see Table 1). Sulfate aerosol is prescribed by two log-normal distributions, which describe Aitken and accumulation modes. Parameters of these two distributions are given in Section 2.5. Following Stenchikov et al. (2021), we assume that the accumulation mode comprises 95% of the sulfate mass, and the Aitken mode 5%.





Mass of ash and sulfate is divided between MOSAIC bins before being passed into the Mie routine. Calculation of mapping coefficients for ash is done by analogy with the method in (Ukhov et al., 2021). The following formula is used to calculate the mapping coefficient $fr_i$ for sulfate aerosol for the $i$th MOSAIC bin with boundaries $r_i$ and $r_{i+1}$ listed in Tab. 2.

$$fr_{(1,2)i} = \frac{\int_{r_i}^{r_{i+1}} r^2 \cdot \frac{1}{\ln \sigma_{1,2} \sqrt{2\pi}} \exp\left(-\frac{(\ln r - \ln r_{1,2})^2}{2\ln^2 \sigma_{1,2}}\right) dr}{\int_0^\infty r^2 \cdot \frac{1}{\ln \sigma_{1,2} \sqrt{2\pi}} \exp\left(-\frac{(\ln r - \ln r_{1,2})^2}{2\ln^2 \sigma_{1,2}}\right) dr}. \tag{29}$$

The computed mapping coefficients for ash and sulfate are presented in Table 3. Ash bin $vash_8$ is only partially accounted for
in $MOS_8$ MOSAIC bin, while the contribution of bin $vash_{10}$ spans across $MOS_1$ and $MOS_7$. We do not include in the Tab. 3 the contributions of the ash bins $vash\_1, 2, ..., 7$ as they are out of the MOSAIC size range and therefore not accounted for in the mass redistribution. Sulfate in Aitken mode only contributes to the $MOS_{1,2,3,4,5}$ bins, $MOS_{3,4,5,6,7,8}$ bins are affected by larger particles belonging to the accumulation mode.

|  | $MOS_1$ | $MOS_2$ | $MOS_3$ | $MOS_4$ | $MOS_5$ | $MOS_6$ | $MOS_7$ | $MOS_8$ |
|---|---|---|---|---|---|---|---|---|
| $vash\_8$ | 0.0 | 0.0 | 0.0 | 0.0 | 0.0 | 0.0 | 0.0 | 0.3561 |
| $vash\_9$ | 0.0 | 0.0 | 0.0 | 0.0 | 0.0 | 0.0 | 0.3561 | 0.6439 |
| $vash\_10$ | 0.1505 | 0.1505 | 0.1505 | 0.1505 | 0.1505 | 0.1505 | 0.0966 | 0.0 |
|  |  |  |  |  |  |  |  |  |
| Sulfate Aitken mode | 0.0003 | 0.0761 | 0.6593 | 0.2607 | 0.0036 | 0.0 | 0.0 | 0.0 |
| Sulfate accumulation mode | 0.0 | 0.0 | 0.0017 | 0.0705 | 0.4336 | 0.4260 | 0.0667 | 0.0015 |

**Table 3.** Ash and sulfate mass redistribution between eight MOSAIC bins.

Aerosol optical properties computed using the volume averaging mixing rule (*aer_op_opt*=1) that assumes internal mixing
of aerosol composition that averages the refractive indices for each MOSAIC bin (Fast et al., 2006). Within each bin, aerosol particles of each species are simplified as spheres undergoing hygroscopic growth. Volume of the water in $i$th bin ($Vol_{h2o,i}$) computed following to Petters and Kreidenweis (2007):

$$Vol_{h2o,i} = \frac{RH}{1-RH} \cdot (Vol_{sulf,i} \cdot \kappa_{sulf} + Vol_{ash,i} \cdot \kappa_{ash}), \tag{30}$$

where $RH$ is the relative humidity, $Vol_{sulf,i}$ and $Vol_{ash,i}$ are the volumes of sulfate and ash, respectively, in the $i$th bin. Ash
hygroscopicity $\kappa_{ash} = 0.1$ (Stuefer et al., 2013), sulfate hygroscopicity $\kappa_{sulf} = 0.5$ (Abdul-Razzak and Ghan, 2004). Aerosol optical depth (AOD) at 300, 400, 600, and 1000 nm as well as SW and LW fluxes at the bottom and top of the atmosphere (BOA and TOA, respectively) are available in the WRF-Chem output. Calculation of volcanic aerosol radiative feedback is activated by setting *aer_ra_feedback* = 1 in the *namelist.input* file.



## 3  Results

Three options are available for simulating volcanic cloud dispersion: volcanic ash with 4 fine or 10 fine and coarse ash species (*chem_opt*=403, *chem_opt*=400), or $SO_2$ with 10 ash species (*chem_opt*=402). The namelist parameter *emiss_opt_vol* defines the constituents of the eruption, i.e. at *emiss_opt_vol*=1 only the emission of ash is simulated, while *emiss_opt_vol*=2 also includes $SO_2$ emission. Here we introduced a new option *emiss_opt_vol*=3 which, besides ash and $SO_2$, also accounts for emissions of sulfate and water vapor. This option requires the emissions to be prepared only using the developed tool

PrepEmisSources (Ukhov and Hoteit, 2025). While *emiss_opt_vol*=1,2 can be used only with emissions prepared using the PREP-CHEM-SRC utility. Hereafter, we use only *emiss_opt_vol*=3 and *chem_opt*=402. The WRF model was configured with the Yonsei University (YSU) planetary boundary layer scheme *bl_pbl_physics*=1 (Hong et al., 2006), the RRTMG longwave and shortwave radiation schemes *ra_lw_physics*=4, *ra_sw_physics*=4 (Iacono et al., 2008), the WSM 5-class microphysics scheme *mp_physics*=4 (Hong et al., 2004), the unified Noah land-surface model *sf_surface_physics*=2 (Tewari, 2004), and the

Grell 3D cumulus parameterization *cu_physics*=5 (Grell and Dévényi, 2002).

### 3.1  Short-term experiments

In order to verify the modifications introduced in the code and to assess the mass balance of ash, sulfate, and $SO_2$, a hypothetical 2-hour long Mt. Pinatubo eruption was modeled, with the total simulation period extending to 30 days (June 15 to July 15, 1991). Domain dimensions are 75x40 grid-cells in zonal and meridional directions, respectively. The domain is centered at

9°N, 95°E. Grid resolution is 100 $km^2$. The meteorological initial and boundary conditions for WRF-Chem are also calculated using the ERA-Interim reanalysis product (Dee et al., 2011) provided at 0.75° × 0.75° horizontal and 6-hr temporal resolution. We emitted 65 Mt of ash, and 15 Mt of $SO_2$. The emissions were redistributed following an umbrella profile spanning from 1 to 15 km, with 95% of the mass contained within the umbrella cloud. The ash fractions for all 10 ash bins were set at 0.1 (refer to *example3.py* in Ukhov and Hoteit (2025)). The prepared emission file is being read every two hours. To avoid the

volcanic debris loss through the boundary, periodic boundary conditions were imposed by setting the following parameters in the *namelist.input*: *specified*=.false., *periodic_x*=.true. and *periodic_y*=.true. .

### 3.1.1  Test of settling velocity of large ash particles

Figure 1 illustrates the settling velocities of ash particles as a function of altitude, with variations in particle radii, calculated both prior to and following the implementation of the correction factor, see Section 2.7.1. Gravitational settling velocity of

sulfate particles having dry volume median radii = 0.62 $\mu$m (see Section 2.5) is also shown. We compare our results with theoretical formulations by Kasten (1968) and Armienti et al. (1988). Presented in Kasten (1968) settling velocities are for spherical particles with radii of 1 $\mu$m, 3 $\mu$m, and 10 $\mu$m. For comparison purposes, the density of these particles was adjusted to align with that of the ash. In Armienti et al. (1988), the ranges of terminal velocities are provided for feldspar mineral particles (with radii of 23.5 $\mu$m, 46.75 $\mu$m, 93.75 $\mu$m, and 187.5 $\mu$m), which are regarded as having the closest density to ash

particles.





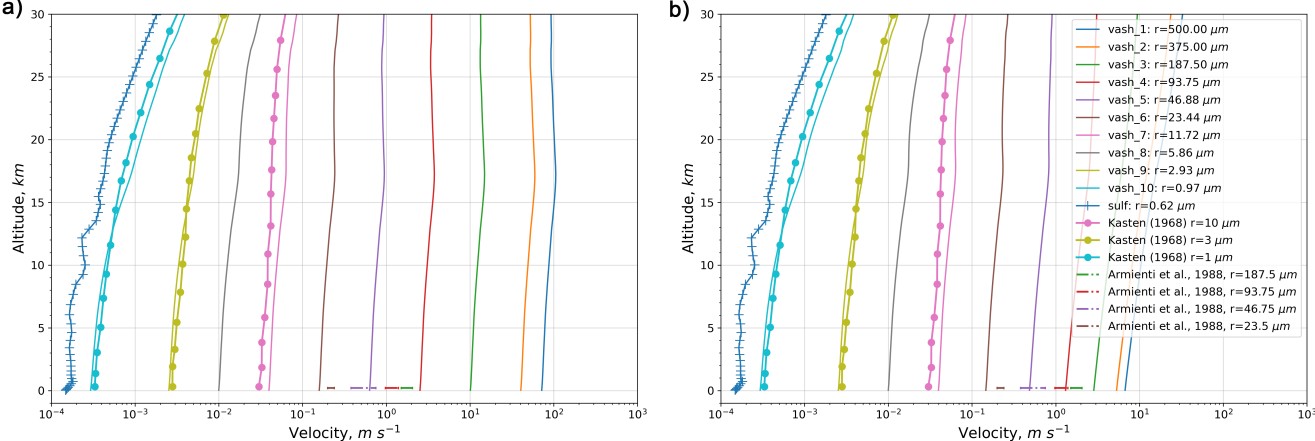

**Figure 1.** Settling velocity as a function of altitude computed for ash particles with different radii and sulfate particle with dry volume median radii = 0.62 μm. Comparison with theoretical data by Kasten (1968) and Armienti et al. (1988). Before a) and after b) correction of the settling velocity of the large ash particles.

Both panels in Fig. 1 show that settling velocities are increasing as a function of particle size and increasing with altitude. Sulfate particles, possessing lower density and smaller size, exhibit the minimal settling velocity. The corresponding curve appears irregular, in contrast to the smoother curves for ash particles, due to the water uptake affecting the size and, therefore, settling velocity of sulfate particles, see eq. (12).

Before the correction, erroneously high settling ($> 40 \mathrm{~m~s}^{-1}$) velocities are observed for the ash particles in the two largest bins. The updated plot (Fig. 1b) shows that the settling velocities of ash particles with radii $\geq 23.44$ $\mu$m decreased across the entire altitude range compared to the original plot (Fig. 1a). This shift is particularly evident for the largest particle classes (e.g., r = 500 $\mu$m, 375 $\mu$m, 187.5 $\mu$m), where the velocity curves move leftward, indicating slower sedimentation. Settling velocity was overestimated up to 10 times for the largest particle bin. After the correction, the velocities of large particles align more closely with those presented in Armienti et al. (1988).

### 3.1.2 Test of ash mass balance

In order to validate the rectified finite-difference scheme applied to the gravitational settling of ash (as described in Section 2.7.2), the model was executed twice: initially without modifications in the scheme and in the settling velocity of large particles, and subsequently with these two changes. In each scenario, the mass balance for ash was computed. Figure 2 illustrates the temporal evolution of the components of the ash mass balance derived from these two simulations. The components are: ash column loading, dry deposited ash, precipitated via large and convective scales ash, gravitationally settled ash, and the cumulative mass (sum of all components). Figure 2a shows that the amount of gravitationally settled ash (green line) increases rapidly, reaching 123 Mt after one month. This dominant sink suggests overactive ash fall. In the rectified experiment (Fig.




2b), only 53 Mt of ash is settled, which is 2.3 times less. Cumulative (loading+depos.+precip) mass (brown line) (Fig. 2a)
shows a continuous increase up to 150 Mt, which significantly exceeds ash emissions (65 Mt), implying a lack of ash mass budget closure with artificial mass gain. In contrast, cumulative mass in Fig. 2b is fixed at 65 Mt, which corresponds to the emitted amount of ash and confirms ash mass conservation. The dry deposition and wet deposition (orange, red, and purple lines) remain minor contributors to the deposition process (Fig. 2a), but they are higher in comparison with the experiment with the corrected scheme (Fig. 2b).

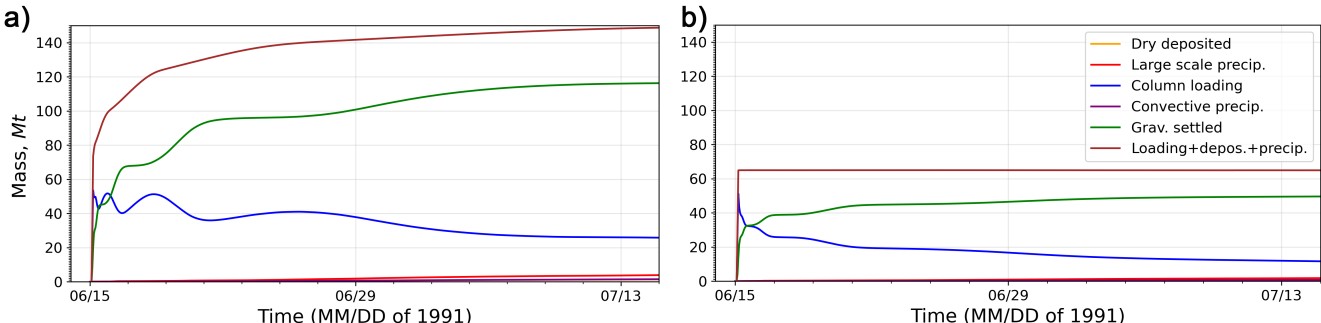

**Figure 2.** Ash mass balance check: before a) and after b) correction of the finite-difference scheme in the gravitational settling subroutine. Deposited ash includes dry deposited ash and gravitationally settled ash. Precipitated ash comprises ash scavenged by large-scale and convective-scale precipitation.





### 3.1.3 Test of SO$_2$ and sulfate mass balance

Figure 3 depicts the mass budget for SO$_2$ and sulfate, assuming 15 Mt of SO$_2$ and 5 Mt of sulfate were initially emitted following the experiment configuration described in Section 3.1. The computed e-folding time for SO$_2$ column loading equals 35.6 days, see Fig. 3a. Oxidation by OH is the dominant SO$_2$ sink, followed by in-cloud oxidation by H$_2$O$_2$. SO$_2$ dry deposition and washout by large-scale precipitation play a secondary role, at least in this configuration of emissions. The sum of all components of the mass balance matches the initial SO$_2$ burden (15 Mt), indicating mass conservation. Mass of sulfate increases from 5 to ≈11.5 Mt by July 15 (Fig. 3b), which indicates continuous formation from SO$_2$ oxidation. 7.5 Mt of sulfate formed via SO$_2$ oxidation, i.e. 5 Mt of SO$_2$ was oxidized (Fig. 3a), which corresponds to 2.5 Mt of sulfur. The amount of formed sulfate is equal to 2.5 Mt of sulfur multiplied by 3 (sulfate molar mass divided by sulfur molar mass), which is equal to 7.5 Mt. Large-scale and convective precipitation are the major sinks for sulfate below the tropopause. A separate experiment with 5 Mt of sulfate emission was run with disabled conversion of SO$_2$ to sulfate (by setting *gaschem_onoff* =0) to validate the sulfate mass balance, see Fig. 3c. Sulfate exhibits a relatively long atmospheric residence time. Only a modest decline (≈10%) of sulfate burden can be observed, primarily due to dry and wet deposition. The cumulative sum of all sinks and remaining burden confirms sulfate mass conservation.

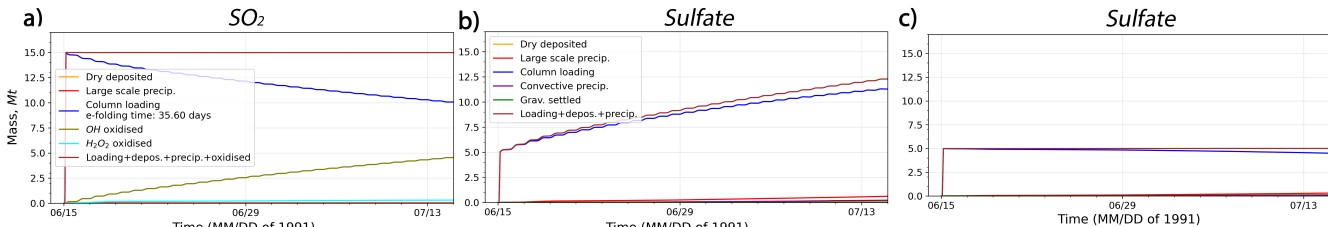

**Figure 3.** Mass budget: a) SO$_2$ at *gaschem_onoff* =1, b) sulfate at *gaschem_onoff* =1, c) sulfate at *gaschem_onoff* =0.





## 3.2 Long-term experiment

We conducted two 3-month simulations of the Mt. Pinatubo eruption at a 100 km grid resolution: CTRL with radiative feedback disabled and PRTB with it enabled. Both simulations start on 15 June of 1991 at 0000 UTC. The eruption starts at 0140 UTC and finishes at 1540 UTC on 15 June of 1991. All sinks for $SO_2$, sulfate, and ash mentioned above are accounted for. The dimensions of the domain are 376x111x60 points in longitude, latitude, and altitude directions, respectively, extending up to 1 hPa. The domain for the simulation is within ≈30°S–60°N latitude belt, see Fig. 8. Periodic boundary conditions on the

longitudinal direction are applied by setting *periodic_x*=.true. in the *namelist.input*.

Emissions of ash and $SO_2$ were prepared using the developed tool PrepEmisSources (see *example2.py* in Ukhov and Hoteit (2025)). We used inverted time-varying emission rates of ash and $SO_2$ from Ukhov et al. (2023), with 66.53 Mt of ash, 15.54 Mt of $SO_2$ in the PRTB run and 62.67 Mt of ash, 16.73 Mt of $SO_2$ in the CTRL run. 75 Mt of water vapor is distributed according to a parabolic profile between 17 and 12 km, the rest 25 Mt linearly between 12 and 1 km. In order to determine the ash mass

fractions for each bin, we follow Stenchikov et al. (2021), assuming that the ash particle size is log-normally distributed with a median radius of 2.4 μm and a geometric width of 1.8. We employed the same method as utilized for sulfate mass redistribution across MOSAIC bins, as referenced in Section 2.8. The computed ash mass fractions are the following: 0.0%, 0.0%, 0.0%, 0.0%, 0.5%, 7.3%, 32.6%, 42.2%, 15.8%, 1.7% for vash_1, vash_2, ..., vash_10 bins, see Tab. 1. In Stenchikov et al. (2021) and Ukhov et al. (2023) we obtained the following mass fractions: 0.1%, 1.5%, 9.5%, 45%, and 43.9% for ash bins 1 to 5,

respectively.

Spectral nudging (Miguez-Macho et al., 2004) has been applied above the PBL (>5.0 km) to horizontal wind components (u and v) for the European Centre for Medium-Range Weather Forecasts Era-Interim reanalysis fields. The nudging coefficient equals $0.0001 \text{ s}^{-1}$. Only wavelengths larger than 450 km are nudged. This setting accounts for the realistic phase of the Quasi-Biennial Oscillation (QBO) and keeps the large-scale motions close to the reanalysis, letting the model develop smaller scale

disturbances freely.

The volcanic cloud after Mt. Pinatubo eruption was observed by several remote sensing instruments, including the total ozone mapping spectrometer (TOMS) (Guo et al., 2004) and stratospheric aerosol and gas experiment (SAGE) (Thomason, 1992). SAGE is a limb-viewing instrument that measures aerosol extinction in the stratosphere at different altitudes. The original SAGE observations have multiple gaps. Thomason (1992) filled these gaps using various techniques. We further refer to this

data set as SAGE/ASAP. Infrared satellite $SO_2$ data provided by the TOVS/HIRS/2 (TIROS (Television Infrared Observation Satellite) Optical Vertical Sounder/High Resolution Infrared Radiation Sounder/2) sensor (Guo et al., 2004).



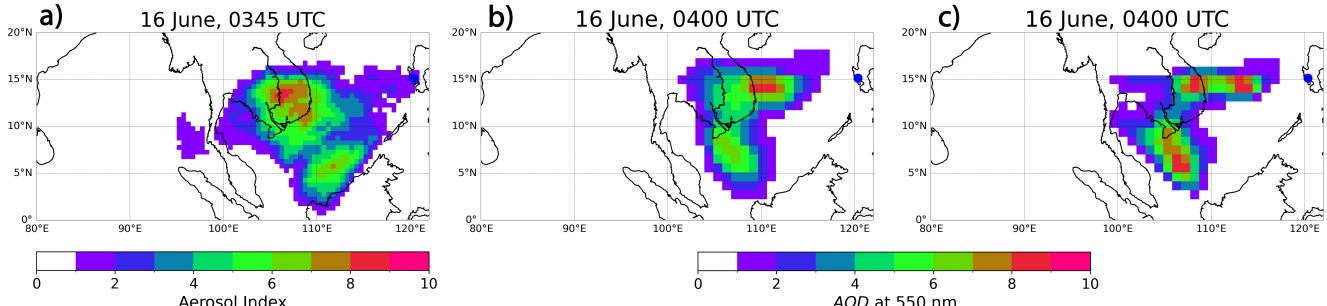

**Figure 4.** Observed on 16 June at 0345 UTC and simulated aerosol diagnostics on 16 June at 0400 UTC: a) TOMS AI; AOD at 550 nm: b) PRTB run, c) CTRL run. Blue dot corresponds to the location of the Mt. Pinatubo.

On the next day after the eruption, the TOMS detected high levels of $SO_2$ loading and positive aerosol index (AI) values, indicating the presence of UV-absorbing volcanic ash. The AI and AOD are linearly related if the volcanic cloud is relatively thin (i.e., AOD < 5) (Krotkov et al., 1999; Ukhov et al., 2023). Figure 4 compares AI from the TOMS retrieval at 0345 UTC on 16 June and AOD at 550 nm computed in the PRTB and CTRL runs at 0400 UTC on 16 June. The spatial patterns of the AI and AOD are not similar for both runs. In Ukhov et al. (2023) a better agreement between these fields was achieved. This discrepancy may be due to different ash bin size ranges and, as a result, different redistribution of ash mass between MOSAIC bins used for the calculation of optical properties, see Section 2.8. However, AOD from the PRTB run slightly better resembles the observed AI field in comparison with the AOD from the CTRL run.



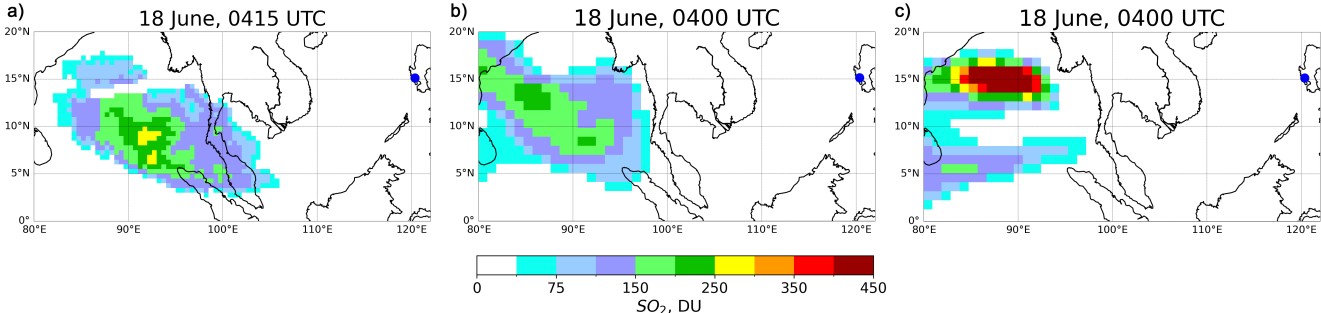

**Figure 5.** Observed and simulated SO$_2$ column loadings (DU) on 18 June. a) the Total Ozone Mapping Spectrometer (TOMS) retrieval; simulated from the (b) PRTB and (c) CTRL runs. 1 DU = $2.687 \times 10^{20}$ (molecules m$^{-2}$). Blue dot corresponds to the location of the Mt. Pinatubo.

Figure 5 compares observed and simulated SO$_2$ column loadings on 18 June, three days after the volcanic eruption. The TOMS observations in Fig. 5a depict a dispersed SO$_2$ plume extending westward from the volcano, with maximum column loadings of approximately 250-300 DU. The plume shows a broad latitudinal spread, with the highest concentrations situated between 5°N and 12°N. The PRTB simulation (Fig. 5b) reproduces the observed westward transport of the SO$_2$ plume more realistically compared to the CTRL run (Fig. 5c). The spatial extent and position of the plume in PRTB align reasonably well with TOMS data, particularly in terms of latitudinal and longitudinal spread and peak loadings. However, the PRTB's plume western and northern extents are slightly overestimated relative to observations. In contrast, the SO$_2$ plume from the CTRL simulation (Figure 5c) displays significant deviations from the observed plume. In particular, the CTRL plume is split into two distinct parts: the northern part with maximum values between 400 and 450 DU, and the southern part with maximum values up to 200 DU. Overall, in contrast with the CTRL run, the PRTB experiment demonstrates an improved match to the TOMS data both in SO$_2$ loading magnitude and spatial distribution.



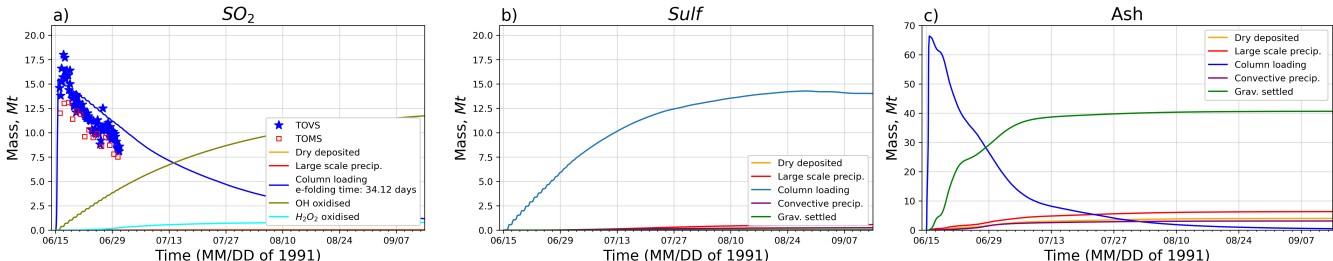

**Figure 6.** Temporal evolution of the mass (Mt) computed for PRTB run: a) $SO_2$, b) sulfate, c) ash along with the contribution of corresponding sinks. TOVS (short for TOVS/HIRS/2) (Guo et al., 2004) and TOMS (Bluth et al., 1992) retrievals of $SO_2$ mass are shown by markers.

Figure 6 presents the temporal evolution of $SO_2$, sulfate, and ash mass during the first 3 months following the eruption, along with the contributions of major sinks. These diagnostics were computed for PRTB run. The initial $SO_2$ mass 15.54 Mt is decreasing steadily over time. This decline is primarily driven by atmospheric oxidation processes converting $SO_2$ into sulfate aerosols, with an effective e-folding time ≈34 days. Minor contributions to $SO_2$ removal arise from dry deposition and

precipitation scavenging, but these represent a small fraction of the overall loss compared to chemical transformation. The simulated rate of decay is close to the TOVS and TOMS estimates.

Sulfate mass increases as $SO_2$ is oxidized primarily by OH, see Fig. 6a. Sulfate mass reached about 14 Mt by the end of August. This growth reflects sustained conversion of $SO_2$, with a minor fraction of sulfate removed by deposition and precipitation. Ash mass dynamics (see Fig. 6c) differ from those of $SO_2$ and sulfate. The ash column loading shows a rapid

initial decline within the first few weeks. Ash mass loss is dominated by gravitational settling and, to a lesser extent, by precipitation and dry deposition.





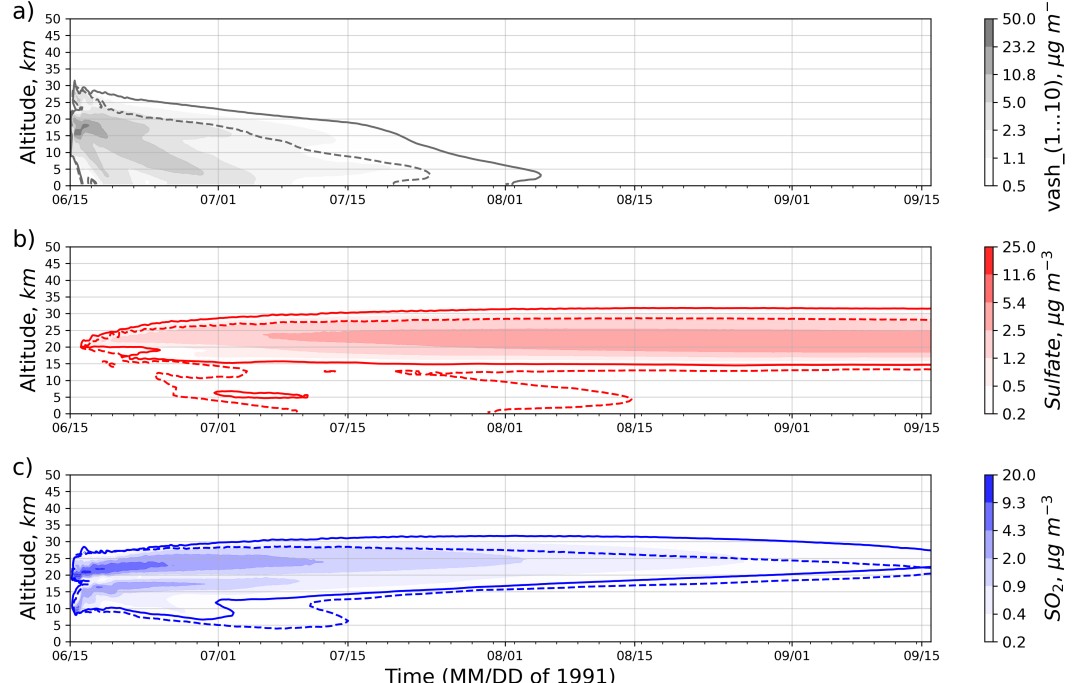

**Figure 7.** Hovmöller diagrams of the temporal and vertical distribution of domain averaged concentrations ($\mu g\ m^{-3}$): a) ash, b) sulfate, c) SO$_2$. Solid contour lines and shading correspond to the PRTB run, which incorporates enabled radiative feedback. Dashed contour lines correspond to the CTRL run.

Figure 7 displays the ash, SO$_2$, and sulfate domain-averaged concentrations as a function of altitude and time computed for the PRTB and CTRL experiments. Solid lines and shading in Figure 7 show the volcanic cloud boundary from the PRTB experiment; dashed lines correspond to the CTRL experiment. The threshold concentrations are at the color bar's lowest value.

In Fig. 7a and 7c, ash and SO$_2$ are present from the beginning of the experiments. It takes about 2-3 days to produce a significant amount of SO$_4$. The PRTB run shows a more pronounced and sustained presence of ash, SO$_2$, and sulfate compared to the CTRL run, suggesting that lofting caused by the radiative heating resists gravitational settling. Despite the gravitational settling, the ash cloud rises initially before descending into the troposphere, see Fig. 7a. The SO$_2$ cloud, driven by buoyancy generated by the radiative heating of eruption products, moves up. In the CTRL run, the stratospheric updrafts also move SO$_2$ and SO$_4$

up, but at a much slower pace. The SO$_4$ cloud's upward motion is restricted by the gravitational settling of sulfate aerosols. Eventually, sulfate deposition velocities define the level of neutral buoyancy for the SO$_4$ cloud. Differential radiative heating and gravitational settling lead to a separation of ash, SO$_2$, and sulfate clouds. In the CTRL simulation, sulfate converted from SO$_2$ below the tropopause undergoes a separation and is rapidly deposited. During the initial two months, the sulfate cloud experiences an ascent; subsequently, it becomes diluted and stabilizes as the buoyancy weakens due to cloud dispersion. In the

CTRL experiment, the SO$_2$ and SO$_4$ clouds end up 3–4 km lower than in the PRTB experiment.



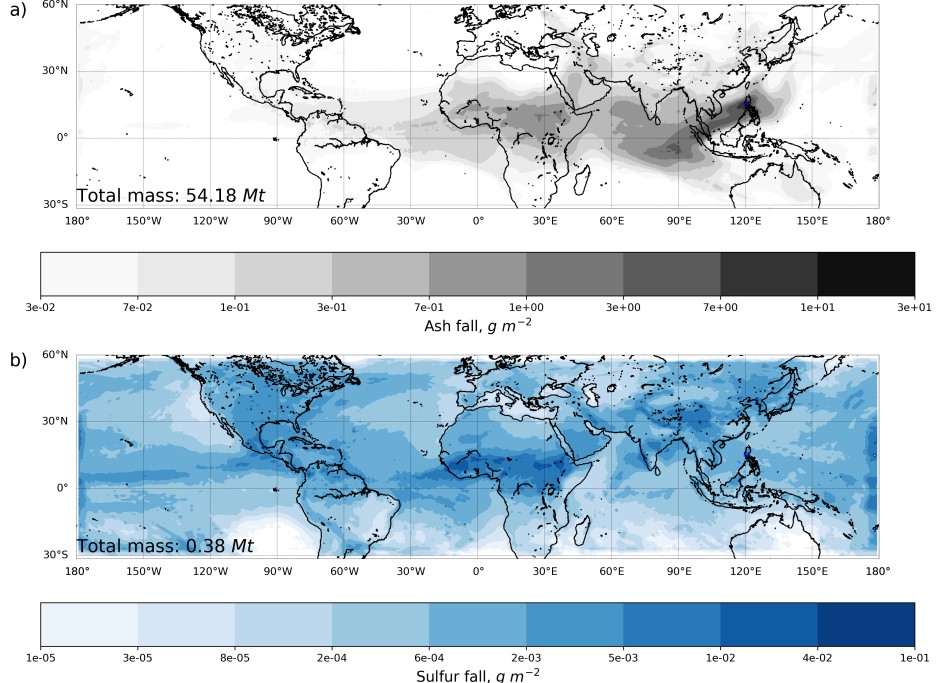

**Figure 8.** Accumulated fall $(\mathrm{g\,m^{-2}})$ over a 3-month period following the eruption: a) ash, b) sulfur. Dry and wet deposition contributions, including gravitational settling, and removal by large-scale and convective precipitation is accounted in both cases.

Figure 8 presents the accumulated deposition of ash and sulfur over a 3-month period following the eruption. The ash and sulfur deposition includes dry deposition, gravitational settling, and washout by large-scale and convective precipitation. The spatial pattern of ash fall is concentrated over Southeast Asia and the western Pacific, where values exceed $1\,\mathrm{g\,m^{-2}}$. In general, spatial distribution of tephra-fall deposits is in agreement with Paladio-Melosantos et al. (1996) and Wiesner et al. (1995). Ash
accumulation also occurred along the equatorial zone in the Indian Ocean, central Africa, and reached the west coast of the Americas, although deposition amounts decreased progressively with distance from the source. Minimal ash fall is seen over the North Atlantic and Central Pacific, as most ash settled within the first few thousand kilometers downwind. The total mass of ash deposited over the domain is 54.18 Mt. This number confirms that nearly all of the injected ash (54.17 out of 66.53 Mt) is removed from the atmosphere within 3 months. The accumulated sulfur deposition sums to 0.38 Mt over the same period.
Given that sulfate mass reached about 14 Mt by the end of August (see Fig. 6b), and that most $SO_2$ had been oxidized, the deposition of 0.38 Mt of sulfur represents only a small fraction of the emitted $SO_2$. The deposition pattern of sulfur is more disperse than that of ash, reflecting the widespread transport, see Fig. 8b. Sulfur fall is generally lower in magnitude, as most of the sulfate aerosol remains suspended at high altitudes and contributes to the long-lived stratospheric sulfate layer rather than being removed by deposition processes. Sulfur deposition occurred broadly across the tropical belt. Noticeable sulfur fall
affected the Pacific Ocean, Central America, and Central Africa. In these areas, we expect the precipitation to be more acidic.



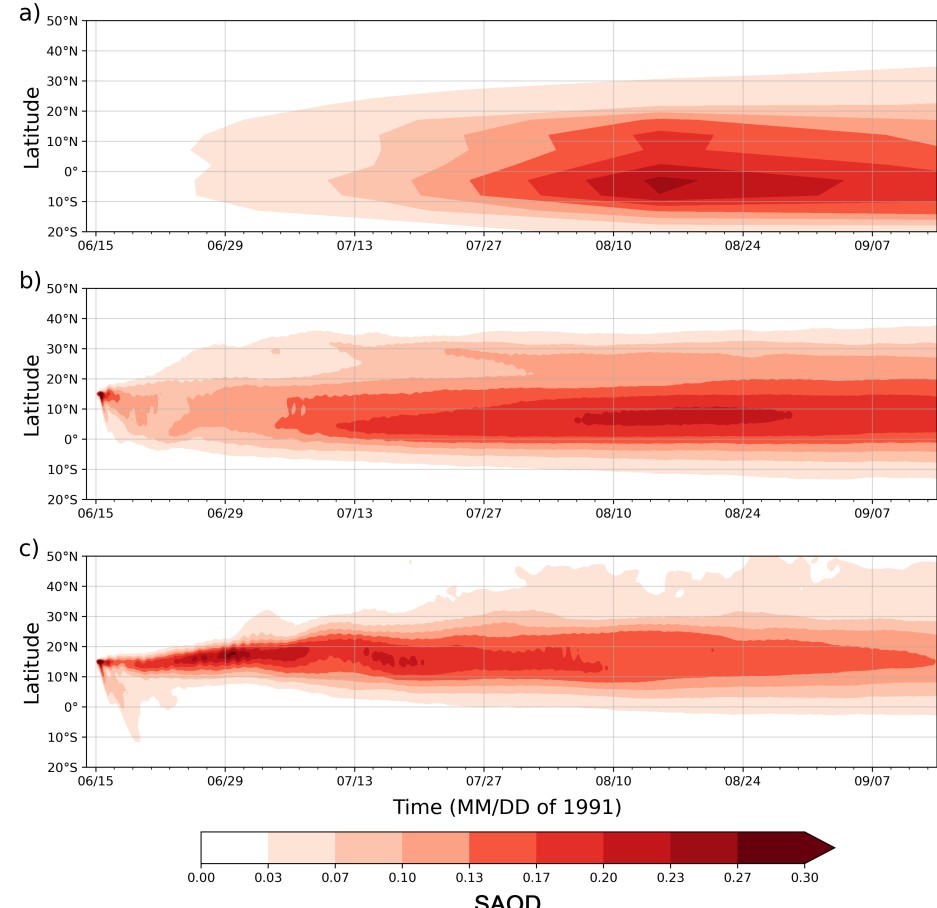

**Figure 9.** Hovmöller diagrams of the zonal mean SAOD: a) the SAGE/ASAP (Thomason, 1992) largely stratospheric AOD at 1.020 µm, b) simulated (stratospheric and tropospheric) SAOD at 1.0 µm from PRTB run, c) same as (b) but from CTRL run.

Figure 9 compares the observed (SAGE/ASAP) (Thomason, 1992) and simulated in PRTB and CTRL runs zonal mean stratospheric aerosol optical depth (SAOD), which includes contributions from ash and sulfate. In the CTRL run, during the first week, a significant amount of aerosol moves to the Southern Hemisphere, reaching 10°S, but the maximum SAOD remains in the Northern Hemisphere. In contrast, the SAOD structure in the PRTB run is close to that observed by SAGE, including the position of maximum SAOD, latitudinal spread, and pattern. But in the PRTB run, the SAOD pattern is shifted by 10° to the north compared with the SAGE/ASAP observations. The maximum SAGE/ASAP SAOD is in the 10°S–0° range, whereas in the PRTB run, the SAOD maximum is within the 0°-10°N range. This suggests that the PRTB simulation slightly overestimates the transport of aerosols into the Northern Hemisphere, which was also observed in our previous study (Stenchikov et al., 2021).



### 3.2.1 Radiative forcing of volcanic aerosol

To estimate perturbations of the Earth's radiative balance, we calculate the change of the total (SW+LW) clear-sky (effects of simulated clouds are ignored) radiative flux $\Delta F$, at the top of the atmosphere (TOA) and at the bottom of the atmosphere (BOA) computed for perturbed (P) and control (C) runs. We define the radiative forcing (RF) as the difference in radiative fluxes:

$$\Delta F_{\text{BOA}} = \left( F_{\text{BOA}}^{\downarrow P} - F_{\text{BOA}}^{\uparrow P} \right) - \left( F_{\text{BOA}}^{\downarrow C} - F_{\text{BOA}}^{\uparrow C} \right), \tag{31}$$

$$\Delta F_{\text{TOA}} = \left( F_{\text{TOA}}^{\downarrow P} - F_{\text{TOA}}^{\uparrow P} \right) - \left( F_{\text{TOA}}^{\downarrow C} - F_{\text{TOA}}^{\uparrow C} \right) = F_{\text{TOA}}^{\uparrow C} - F_{\text{TOA}}^{\uparrow P}, \tag{32}$$

where $\downarrow$ denotes a downward flux and $\uparrow$ an upward flux. P refers to the PRTB run with activated aerosol radiative feedback, and C refers to the CTRL run with disabled radiative feedback. This notation assumes that positive flux values indicate a warming effect, while negative values indicate a cooling effect. 2D fields of the SW and LW clear sky fluxes at the TOA and BOA are available in the WRF-Chem output. Zonal averaged clear sky RFs at TOA and BOA are shown in Figure 10, and their three-month and domain-averaged values are summarized in Table 4.

Volcanic aerosols heat both the TOA/BOA by increasing the upward/downward LW radiation from the aerosol cloud. But atmospheric absorption substantially decreases the LW perturbations at the BOA. In particular, the domain-averaged LW RF is stronger/weaker at TOA/BOA, 1.5/0.2 W m$^{-2}$, respectively. Strong SW cooling dominates at TOA and BOA, domain-averaged

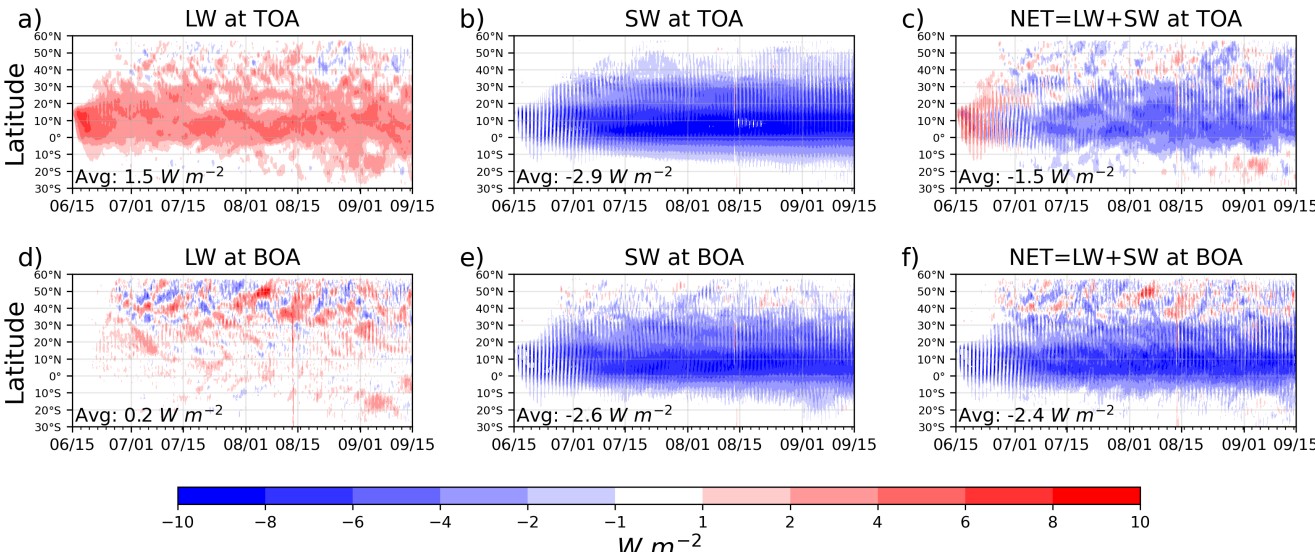

**Figure 10.** Hovmöller diagrams of the zonal averaged clear sky RFs for SW, LW, and NET=SW+LW (W m$^{-2}$) at TOA and BOA. Domain-averaged values of RFs are shown for each panel at the bottom right corner.



|  | LW Avg, (W m$^{-2}$) | SW Avg, (W m$^{-2}$) | NET=LW+SW Avg, (W m$^{-2}$) | NET=LW+SW Global Avg, (W m$^{-2}$) |
|---|---|---|---|---|
| TOA | 1.5 | -2.9 | -1.5 | -1.0 |
| BOA | 0.2 | -2.6 | -2.4 | -1.6 |
| TOA-BOA | 1.3 | -0.3 | 0.9 | 0.6 |

**Table 4.** Radiative forcings for LW, SW, and NET=LW+SW (W m$^{-2}$) at TOA and BOA three-month averaged over the simulation domain and over the globe.

values are -2.9 W m$^{-2}$ and -2.6 W m$^{-2}$, respectively. SW cooling at the TOA is caused by sulfate aerosol efficient scattering of solar radiation back to space. SW cooling at the BOA is conditioned by the reduced downward SW flux.

Short-lived ash particles and sulfate aerosol absorb some outgoing LW radiation, which causes warming at TOA. Simultaneously, ash and sulfate also scatter incoming SW radiation, contributing to a cooling effect, but due to the ash presence,
warming at the TOA is stronger in the first two weeks. Therefore, during this period, ash dominates the NET=LW+SW RF at TOA, inducing a warming at the TOA and counteracting SW cooling. After ash fallout, sulfate's SW cooling prevails over LW warming at the TOA. Calculated domain average value for NET RF at the TOA is -1.5 W m$^{-2}$, see Fig. 10c.

SW cooling prevails over LW warming at the BOA as well; domain-averaged values are -2.6 and 0.2 W m$^{-2}$, respectively. Domain average value for NET RF at the BOA is -2.4 W m$^{-2}$. Over the large areas in the equatorial zone, NET cooling at the
BOA reaches -6 to -8 W m$^{-2}$, see Fig. 10f. These values of RFs are in good agreement with other modeling studies (Stenchikov et al., 1998; Ramachandran et al., 2000).

The simulation domain covers approximately 68.3% of the globe. Therefore, to compute the global average RFs, we multiply the domain average RF value by 0.683, see the right column of Table 4. Thus, the global average NET=LW+SW clear-sky RF is -1.0 W m$^{-2}$ and -1.6 W m$^{-2}$ at TOA, BOA, respectively.
Difference of the NET fluxes at the TOA and BOA reflects the change of the total radiative balance of the whole atmospheric column, where a positive difference of TOA and BOA (domain and global average values are 0.9 and 0.6 W m$^{-2}$, respectively) represents the heating of the part of the atmospheric column, where the sulfate aerosol cloud resides. Again, this heating is driven by the LW adsorption by sulfate aerosols.



## 4    Conclusions

In this study, we enhanced the WRF-Chem v4.8 model by implementing physical and chemical mechanisms that are necessary to simulate the realistic evolution of the volcanic clouds in the atmosphere. In particular, we corrected the derivation of the finite difference scheme for gravitational settling of the ash. Specifically, ash mass balance was strongly violated in the previous WRF-Chem model runs. To the authors' knowledge, the found inconsistency, despite the long-term usage, has not previously been recognized nor reported. We also introduced the correction factor for the deposition velocity for coarse ash particles

(radii $\geq 23.44$ μm). We refined $SO_2$ oxidation processes and implemented gravitational settling for sulfate aerosols, which is important for volcanic stratospheric clouds and was previously neglected. We implemented an interaction of ash and sulfate aerosols with SW and LW radiation. This so-called direct radiative effect of aerosols refers to the radiation changes caused by aerosol absorption and scattering. In general, our modifications and additions provide a more physically consistent representation of volcanic plume dynamics and will improve volcanic ash and $SO_2$ forecasts, benefiting both scientific research and

operational applications. Additionally, we developed an open-source preprocessor called PrepEmisSources (Ukhov and Hoteit, 2025), which facilitates the preparation of the volcanic emission file used by WRF-Chem, and emissions can be time varying. For this, we introduced a new option *emiss_opt_vol*=3 which, besides ash and $SO_2$, also accounts for emissions of sulfate and water vapor. All these new capabilities are available under *chem_opt*=402 option in the *namelist.input* file. Therefore, we encourage using the PrepEmisSources utility along with *emiss_opt_vol*=3, which can be used with *chem_opt*=402, 400, or 403.

We demonstrate the effect of changes implemented into the WRF-Chem v4.8 model by running one short-term and two long-term experiments (with enabled and disabled radiative feedback, respectively), which simulate the Mt. Pinatubo eruption. In all experiments, the emissions were prepared using the developed PrepEmisSources utility. In the former run, we successfully established the mass balance for ash, $SO_2$, and sulfate accounting for all accounted major sinks by prescribing periodic boundary conditions to avoid mass loss through the lateral domain boundaries. Using the long-term runs, we estimated the

role of the radiative heating, traced the spatio-temporal evolution of the AOD and distributions of the ash and sulfate clouds, and calculated the maps of ash and sulfur fallout. Where possible, the results of the simulations were compared with the observations.

We found that the radiative effect of eruption products improves the model's ability to predict the transport of volcanic clouds and increases their persistence in the atmosphere. Interaction of the aerosols with atmospheric radiation through scattering and

absorption was also significant and resulted in cooler surface temperatures. In particular, in the equatorial area, the net cooling at the BOA reaches -6 to -8 W m$^{-2}$, which is within the acknowledged range shown in other studies. In the long-term experiment, the model captured the transition from ash-dominated to sulfate-dominated forcing at the TOA. Just after the eruption, the ash radiative forcing dominates over the weak sulfate radiative forcing, which strengthens after two weeks when enough sulfate mass has formed.

We demonstrated that the enhanced WRF-Chem v4.8 model could be useful in multiple applications. Starting from the simulation of the impact of volcanic clouds on aviation and forecasting ash and sulfur fallout, and finishing by calculating the



volcanic cloud's effect on climate. We hope that new additions and rigorous validation provided in this paper could help to promote the WRF-Chem v4.8 to the cohort of the VATD models used by VAACs.

This work is in line with the open-source paradigm and will help WRF-Chem users to better handle the code and understand
physical interconnections. In the course of the paper, we tried to designate places in the code where we implemented changes and where the model parameters can be changed. As the adjustments for the specific eruption might need to be made. Firstly: ash size distribution, refractive index, and ash density. Secondly, sulfate size distribution parameters for both modes.

## 5   Future work

Recently, one of the directions of geoengineering modeling, such as Stratospheric Aerosol Injection (SAI) has become popular.
The idea behind SAI is mimicking the cooling effects of volcanic eruptions by injecting aerosols into the stratosphere, where they would scatter some incoming shortwave radiation and cool the Earth's surface. Recent research (Stefanetti et al., 2024) suggests that solid particles, such as diamond, could be used as these aerosol particles. It was shown that diamond is most efficient in reducing global warming per unit injection. In contrast to sulfate aerosols, it has fewer side effects, such as less absorption of terrestrial infrared radiation, which results in stratospheric warming and reduced cooling efficiency. In addition,
diamond is a chemically inert material, which does not cause ozone depletion. We would like to note that in its current configuration, the enhanced WRF-Chem v4.8 model could be potentially used for the simulation of SAI and studying its effects on climate, at least using diamonds as injected material. Inclusion into the model calculation of the heating rates by analogy with the Stenchikov et al. (2021) would provide a better view on the vertical redistribution of cooling and warming within the atmospheric column. Adding the "double call" method (Stenchikov et al., 1998) into the radiation calculation would allow for
the separate calculation of the radiative forcings for ash, sulfate, and water vapor (Stenchikov et al., 2025). The next step in further enhancing the code would be adding a relevant chemical mechanism using the kinetic preprocessor (KPP) (Damian et al., 2002) for the simulation of stratospheric or tropospheric chemical reactions, including the photolysis reactions. Adding volcanic ash aggregation (Lemus et al., 2025) would also improve the agreement with the observations of ash fallout. Owing to its modular design and flexibility, the developed PrepEmisSources code can be easily extended to simulate emissions of
different types. For example, emissions caused by industrial or wildfires.

## Appendix A:  PrepEmisSources utility

The default methodology for preparing volcanic emissions file comprised of two distinct stages. Initially, it is necessary to employ an open-source tool, PREP-CHEM-SRC (Freitas et al., 2011), with a limited set of options pertinent to the parameters governing the eruption itself. In the subsequent stage, a utility program *convert_emiss.exe* embedded within the WRF-Chem
code must be executed. This utility program reads the intermediate binary data file generated by PREP-CHEM-SRC and computes the vertical mass distribution and the emissions for the volcanic ash and $SO_2$. Then, computed arrays are stored in the WRF-Chem emission file. During the WRF-Chem's runtime, the emission file is processed a single time at the onset of





the simulation; consequently, it becomes impossible to delineate emissions that vary temporally. Moreover, the dispersion of emissions vertically is restricted exclusively to a 75/25 umbrella-shaped plume, where 25% of the mass is distributed from vent

height to a $\approx$ 73% of plume height, while 75% follows a parabolic distribution to plume-top height.

In order to streamline and refine the preparation of volcanic emissions, we have developed the PrepEmisSources utility. This tool, implemented in Python, is specifically designed to prepare and visualize volcanic emission scenarios for integration with atmospheric models such as WRF-Chem. The utility enables the construction of 4D (space-time-altitude) emission profiles with configurable spatial, temporal, and vertical resolutions. It supports multiple emission types (e.g., ash, $SO_2$, sulfate, water

vapor) and vertical distribution types (e.g., uniform, umbrella, Suzuki (Suzuki et al., 1983; Mastin and Van Eaton, 2020)), and can accommodate both synthetic and inversion-derived emission scenarios (Ukhov et al., 2023; Brodtkorb et al., 2024). With its object-oriented and extensible architecture, PrepEmisSources allows for flexible scenario definition and integration of external data. Emission profiles are exported to NetCDF files in a format directly compatible with WRF-Chem input requirements, facilitating seamless simulation of volcanic events. Visualization tools are included for diagnostic analysis of emission

structures before model execution.

To accommodate the capability to read the emissions at specific intervals, we implemented changes in the WRF-Chem's logic of reading the emission file via auxiliary input 13. In particular, emission file will be read by the WRF-Chem at uniform time intervals (namelist parameter *auxinput13_interval_m* in minutes) and a prescribed number of times (namelist parameter *frames_per_auxinput13*). This option works when *chem_opt*=400, 402 or 403 and *emis_opt_vol*=3. More details on how to use

the PrepEmisSources utility are presented in Ukhov and Hoteit (2025).



## Appendix B: List of output diagnostics

|  | **Process** | **namelist option** | **Output field name** | **Units** |
|---|---|---|---|---|
| **Ash** | Dry deposition | *vertmix_onoff*=1 | ASH_DRYDEP | kg m$^{-2}$ |
|  | Grav. settling | *vertmix_onoff*=1 | ASH_FALL | kg m$^{-2}$ |
|  | Large scale scavenging | **wetscav_onoff*=-1 | WD_ASH_SC | g m$^{-2}$ |
|  | Conv. scale precip. scav. | *conv_tr_wetscav*=1 | WD_ASH_CU | μg m$^{-2}$ |
|  | Dry deposition | *vertmix_onoff*=1 | SULF_DRYDEP | mol m$^{-2}$ |
| **Sulfate** | Grav. settling | *vertmix_onoff*=1 | SULF_GRAV_SETL | kg m$^{-2}$ |
|  | Large scale scav. | **wetscav_onoff*=-1 | WD_SULF_SC | mmol m$^{-2}$ |
|  | Conv. scale precip. scav. | *conv_tr_wetscav*=1 | WD_SULF_CU | mmol m$^{-2}$ |
|  | Dry deposition | *vertmix_onoff*=1 | SO2_DRYDEP | mol m$^{-2}$ |
| **SO$_2$** | Large scale scav. | **wetscav_onoff*=-1 | WD_SO2_SC | mmol m$^{-2}$ |
|  | Oxidation by H$_2$O$_2$ | *gaschem_onoff*=1 | SO2_H$_2$O$_2$_LOSS | kg of Sulfur |
|  | Oxidation by OH | *gaschem_onoff*=1 | SO2_OH_LOSS | kg of Sulfur |
| * does not work with *cu_physics*=3 or 10. In our case *cu_physics*=5 | | | | |

**Table B1.** List of output diagnostics and corresponding namelist options.



*Author contributions.* A. Ukhov planned and performed the calculations, wrote the manuscript, and led the discussion. A. Ukhov and J. Schnell implemented WRF-Chem code modifications and additions. All authors participated in the discussion and reviewed the manuscript.

*Competing interests.* The authors declare that they have no conflict of interest.

*Disclaimer.* The statements, findings, conclusions, and recommendations are those of the author(s) and do not necessarily reflect the views of NOAA or the U.S. Department of Commerce.

*Acknowledgements.* The research reported in this publication was supported by funding from King Abdullah University of Science and Technology (KAUST). For computer time, this research used Shaheen III managed by the Supercomputing Core Laboratory at King Abdullah University of Science & Technology (KAUST) in Thuwal, Saudi Arabia. The research was also supported in part by the NOAA cooperative
agreement NA22OAR4320151, for the Cooperative Institute for Earth System Research and Data Science (CIESRDS).

*Code and data availability.* The latest version of the WRF-Chem model is available at https://github.com/wrf-model/WRF. The WRF-Chem code used in this publication along with namelist files and scripts for OH and $H_2O_2$ interpolation are archived at https://doi.org/10.5281/zenodo.16894619 (Ukhov, 2025). The PrepEmisSources utility is archived at https://doi.org/10.5281/zenodo.16856541 (Ukhov and Hoteit, 2025) is also available at https://github.com/saneku/PrepEmisSources.



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
