# Peer review of "Enhancing Volcanic Eruption Simulations with the WRF-Chem v4.8"

_EGUsphere, 2025_

## Author Comment (AC1)

Review of GMD paper "Enhancing Volcanic Eruption Simulations with the WRF-Chem v4.8"

The paper "Enhancing Volcanic Eruption Simulations with the WRF-Chem v4.8" describes a few updates in WRF-chem to improve the volcanic ash and sulfur simulations. The author well described the changes they made, showing figures of parameters improvement after the changes, and apply the real example of Mt. Pinatubo as a test case. The paper is pretty well-rewritten and I recommend the publication after the author addresses the major comments I listed below. This is because as a GMD paper, I expect a little more clear and thorough description the model itself.

We would like to thank the reviewer for his thorough and positive feedback. Below, we address his comments. Our replies are provided in blue. Changes in the text are in *italic*.

**Major:**

1. Between section 1 previous work and section 2 code modification

I highly recommend to add a section here for model description. You described so many previous works using WRF-chem, but apparently not all the modifications are used in your current version. It is quite confusing not knowing your start point. So, I recommend you to add and additional section with the following three aspects:

Only describe the modification cannot comprehensively for the audience to know the whole structure. You need to at least briefly describe the sulfur chemistry and microphysical processes in WRF-chem for how the sulfate forms, how does sulfate interact with dust (or there's no interactions, then you should quote some of these interactions may be important for future development for initial plume dispersion), how does the nucleation and coagulation work to convert H2SO4 to sulfate, how does the aerosols interact with the cloud.

It's hard to know what processes you consider besides the processes you modified. For example, how are you consider the formation of ash aggregates (Brown et al., 2012), sulfate nucleation (what sulfate nucleation scheme you use?) and coagulation (is Van De Var force considered? is sulfate coagulate with ash?), removal of sulfate due to adsorption on ash particle (Zhu et al., 2020). These are all very important processes for calculating removal from atmosphere, fall velocity and sedimentation. If they are previously implemented, you need to describe them. If they are not considered in your current version, you should mention some processes are not currently considered and acknowledge their importance.

A schematic diagram will be really helpful to show the workflow of the model, writing down what is the code before and use a different color to show what has been implemented by this work.

We appreciate the reviewer's suggestion to clarify our 'start point'. To improve clarity, we renamed and split Section 1.1 into two parts: "1.1 Previous and related work" and "1.2 Objectives and implemented improvements." This restructuring clearly distinguishes prior WRF-Chem developments from the modifications introduced in our study.

We acknowledge that a comprehensive description of sulfur chemistry and microphysical processes would substantially expand the manuscript beyond its technical focus. These processes are either simplified or not explicitly represented in our modeling framework and are already documented in the literature. We provided sufficient context for the reader to understand the capabilities of the proposed model framework while keeping the paper focused on the newly implemented developments.

The following text was added into the 'Future work' section: "Ash aggregation, sulfate nucleation, interactions between sulfate and ash, and heterogeneous sulfate removal via adsorption on ash surfaces are not accounted for in this version. Inclusion of these processes into the model would be a natural extension of this work."

We believe that Sections 1.2 and 2 clearly communicate the 'start point' and the implemented modifications. Therefore, adding a schematic diagram would largely replicate information already conveyed in the text. Consequently, we would like to keep it as it is.

2. Section 2.5 The description of sulfate size distribution implementation is not easy to follow. Also, some implication is questionable:

Section 2.5: Now, you are saying you use a size distribution for aerosol. could GOCART track the size distribution? It should just track the number and total burden, right? since it is a two-moment scheme.

Yes, we use a bulk (single-moment) approach to represent sulfate aerosols. In this approach, only the mass mixing ratio is prognosed, while the size distribution is prescribed and fixed. Accordingly, we have revised the first sentence of Section 2.5: "As in the GOCART aerosol module, a bulk (single moment) approach is used to represent sulfate aerosols. In this approach, only the mass mixing ratio is tracked and the assumed size distribution is fixed."

Line 168: "0.2 um" radius or diameter? Are you talking about background sulfate or volcanic sulfate aerosol?

This value refers to volcanic sulfate aerosol, and it represents the particle diameter. The text has been updated accordingly to clarify this: "Volcanic sulfate aerosol droplets in the troposphere have diameters smaller than 0.2  $\mu$ m. Therefore, in the troposphere, only dry and wet scavenging of sulfate aerosols is usually considered."

Line 169: "bigger", The stratospheric background sulfate usually not bigger than 0.2 um radius. but volcanic sulfate is bigger. So again, it is a little confusing what you are talking about.

It is about volcanic sulfate. The text has been updated as follows: "In contrast, in the stratosphere, the **volcanic** sulfate aerosol droplets are bigger, air density is lower, and gravitational settling becomes the leading deposition process."

Line 173-174: is the size distribution for volcanic sulfate too or just background sulfate? The size distribution between them is very different.

The text has been updated as follows: "... we assume that the **volcanic** sulfate aerosol number-density size..."

Eqn 12: my impression is that kappa values are suitable for tropospheric aerosols but too low for the stratospheric sulfate. You should make a plot of H2SO4 weight percent inside the H2SO4-H2O binary solution as a function of altitude. The stratospheric H2SO4 weight percent generally ranges from 60-90%.

Following the comment, we produced a plot of the RH profile (Figure A) and H2SO4-H20 solution weight percent as a function of altitude for  $\kappa$  = 0.5, 1.19, and 1.4. As shown in Figure b), the resulting H2SO4 weight is reaching 60–90% in the stratosphere across selected  $\kappa$  values, which agrees with the

reviewer's comment. Lower  $\kappa$  values yield higher acid concentrations due to reduced water uptake, while higher  $\kappa$  values lead to more diluted solutions.

Eqn 30: You previously listed the hygroscopicity of sulfate with a kappa of 1.19 in Eqn 12, but here you list the sulfate kappa of 0.5.

Thank you for noting that. We have added a new paragraph to explain this difference: "Following Aquila et al. (2012), we assume a hygroscopicity parameter of  $\kappa$  = 1.19 for sulfate particles, which contrasts with  $\kappa$  = 0.5 used in the calculation of sulfate optical properties (see Section 2.8). We found that  $\kappa$  = 0.5 produced unrealistically small Rwet, sulf values, resulting in slower gravitational settling and artificial accumulation of sulfate in the troposphere. In contrast, using  $\kappa$  = 1.19 increased Rwet, sulf, and, consequently, the gravitational deposition velocity, yields a more realistic vertical distribution and removal rate of sulfate aerosols. Furthermore, we verified that  $\kappa$  = 1.19 results in an H2SO4–H2O binary solution weight fraction in the range of 60–90%, consistent with observational estimates for stratospheric sulfate aerosols."

To illustrate this effect, we plotted the *Rwet,sulf* vs Altitude in Figure C above. The Python script for plotting is provided at the end of the document.

**Minor:**

Line 14: and chemical transformation-> and SO2 chemical transformation

**Updated as suggested.**

Line 37: "In contrast, water vapor and sulfate are important in further stages of volcanic cloud dispersion."->"In contrast, sulfate, and sometimes water vapor, are important in further stages of volcanic cloud dispersion."

**Corrected as suggested.**

Line 50: "In the model, ash is gravitationally settled, and SO2 undergoes oxidation to sulfate using the prescribed OH vertical distribution." After reading the whole paper, I am still not sure if you use the same prescribed OH as they are. I thought you changed the OH to be interactive, but from you description later

on line 106-109, it seems to be not. So please specify if you use the same method as they are, or if you improved, what's the difference between yours and theirs.

As in the previous version, the updated model uses prescribed OH fields rather than interactive chemistry. Our approach does not include ozone photolysis or interactive OH production, since developing a fully coupled chemical system was beyond the scope of this study. Specifically, OH concentrations are prescribed from the CLaMS model output. In addition, we updated the reaction rate coefficients for the SO2+OH oxidation and implemented in-cloud SO2 oxidation by H2O2, which were not included in the previous version.

Line 106-109: If they are not interactive, how could you deplete OH? Are you having an OH fiend vary with time because the sun rise and sun set? But SO2 doesn't deplete OH?

The OH field is prescribed, and the SO2 does not deplete OH. The solar cycle is accounted for. This is mentioned in Line 120. We added one word, 'prescribed', to the sentence that starts on Line 121. This sentence now reads as follows: "To address this variability, the prescribed concentration of OH is multiplied by a scaling factor, which depends on the solar zenith angle."

Line 53-55, Line 61-64: these sentences just describe what the previous work did, but haven't list what they found. You other descriptions in this section are good.

We thank the reviewer for this observation. We have revised Lines 53–55 and 61–64 to include summaries of the main results of those studies. The text (Lines 53–55) reads now as follows: "Webley et al. (2012) provided a detailed evaluation of the 2010 Eyjafjallajökull eruption in Iceland and successfully reproduced the observed plume structure. Steensen et al. (2013) conducted a qualitative comparison of the 2009 Mount Redoubt volcanic clouds using the PUFF and WRF-Chem models together with satellite observations, showing that WRF-Chem and PUFF produced comparable ash transport patterns."

For lines 61-64: "Rizza et al. (2023) simulated a sequence of Mt. Etna paroxysms by coupling WRF-Chem with near-source L-band radar observations and demonstrated improved agreement between modeled and observed plume heights. Egan et al. (2020) analyzed tephra fallout and in situ airborne measurements of ash from the 2010 Eyjafjallajökull eruption, providing valuable validation data for volcanic ash dispersion modeling. de Bem et al. (2024) employed WRF-Chem to simulate ash transport from the 2015 Calbuco eruption in Chile and successfully reproduced the observed regional dispersion and deposition patterns."

Line 71-73: I am curious why you use "MOSAIC" instead of "GOCART" for this research, because MOSAIC is generally for tropospheric chemistry especially near the surface, rather than upper atmosphere.

We agree with the reviewer's observation. The sentence has been removed from the revised manuscript.

Line 87-89: This is a following question for line 71-73. Stenchikov et al., 2025 is using MOSAIC in the Hunga case, right? But you are using GOCART. Do you need to apply some extra work to add the water vapor and sulfate in since you are using different aerosol schemes?

Indeed, Stenchikov et al. (2025) employed the MOSAIC aerosol scheme, while our study did not use GOCART directly. Instead, we developed our own volcanic aerosol scheme, in which the SO2 oxidation by OH and H2O2 follows the GOCART formulation. Other parameterizations, such as dry and wet deposition and gravitational settling, were adapted and extended from the GOCART scheme. Therefore, our implementation represents an independent, tailored scheme specifically designed for volcanic applications. All modifications and improvements are presented in the last paragraph of Section 1.

Line 138: "large scale scavenging" Is it stratiform clouds scavenging? you should add a little explanation.

Yes, the term 'Large-scale scavenging' refers to scavenging within stratiform (non-convective) clouds. We have added the following text to the beginning of Section 2.3 to clarify this: "Here, large-scale scavenging represents in-cloud removal processes associated with stratiform clouds, as opposed to convective precipitation."

Line 186: you should describe the theories you use to calculate the fall velocity of sulfate and ash. Cite the paper.

The theoretical formulations and references used to calculate the fall velocities of sulfate and ash are already provided in Sections 2.5 and 2.7.1. Specifically, sulfate settling follows Stenchikov et al. (2021). It uses the hygroscopic growth parameterization of Petters and Kreidenweis (2007), while ash settling employs the Stokes–Cunningham framework with large-particle drag correction (Mailler et al. 2023). We believe that the theoretical framework for calculating the fall velocities of sulfate and ash is already presented and cited comprehensively in the manuscript.

Line 205 and 211: μu -> μm?

**Corrected.**

Line 234-235: "where air density is less by two orders of magnitude with respect to the surface" only one order of magnitude if you talk about UTLS where most volcano injected into.

Corrected. Now, the sentence reads as follows: "Ash is usually emitted into high altitudes, where air density is less than on the surface."

Line 273-275 and the whole document where you say "MOSAIC bin": It's really confusing to mention MOSAIC here since you are not using MOSAIC (you are using GOCART), but just have identical bin width as MOSAIC. Any model can define a bin width. I recommend to change "MOSAIC bins" to a different name.

We would like to keep this naming for consistency with our other published papers.

Question: These bins don't cover Aitken mode and large ash particles. Any shortcomes?

The MOSAIC bins used in this study cover both the Aitken mode and the optically relevant ash size range. As shown in Table 3, we present the ash and sulfate mass mapping into MOSAIC bins. This mapping explicitly assigns Aitken-mode sulfate mass mainly to MOS3–MOS4, with smaller fractions in MOS1, MOS2, and MOS5. At the other end, the largest ash bins (vash\_1,..,7) are outside the optical range by design, as they are dynamically treated through gravitational settling and deposition but contribute negligibly to the aerosol optical depth due to their short atmospheric lifetime. Therefore, the MOSAIC configuration adequately represents all radiatively and dynamically relevant particle sizes.

Line 301: "10 ash species" do you mean 10 sulfate species?

We indeed mean 10 ash species, corresponding to the 10 ash particle size bins defined in the model, rather than 10 sulfate species.

Line 317: "We emitted 65 Mt of ash, and 15 Mt of SO2." What these numbers based on? Citation needed.

As stated at the beginning of Section 3.1, this experiment represents a hypothetical two-hour eruption at the location of Mt. Pinatubo. This short-term experiment was designed to verify the code modifications and assess the mass balance of ash, sulfate, and SO2 in a short-term test run. Therefore, the emission magnitudes of 65 Mt of ash and 15 Mt of SO2 were chosen arbitrarily for testing and plotting convenience, rather than based on specific values. For the long-term experiment described in Section 3.2, we used more realistic emission estimates derived from Ukhov et al. (2023).

Line 318: "The ash fractions for all 10 ash bins were set at 0.1" What does this mean?

We have rewritten this sentence as follows: "The total emitted ash mass was evenly distributed among the ten ash size bins, with each bin assigned 10% (fraction = 0.1) of the total ash mass."

Figure 1: move the legend outside the panel b

**Corrected.**

Figure 3: b and c are not easy to understand. I thought column loading should be the highest line because it only contains production term. Then, other line applies one or several sink terms. So why the grey line is the highest? where's the green line?

Column loading represents the total amount of material remaining in the atmosphere. It would be the highest curve only in the absence of removal processes. In our case, however, dry deposition, large-scale and convective precipitation, and gravitational settling continuously reduce the atmospheric burden. There is no grey line in the figure. We assume that the reviewer is referring to the brown line, which represents the sum of the column loading and all deposited material (dry and wet). Some lines (including green, which corresponds to gravitational settling) may not be visible because certain removal processes have zero or near-zero values in this experiment. The relative contributions of individual processes are shown more clearly in Figure 6.

Why dry deposition and gravitational settling are two different lines (they are the same physical process, just one in the air and one on the surface, right)?

Yes, they are different processes. To clarify this, we added the following text into Section 2: "We distinguish between gravitational settling and dry deposition processes. Gravitational settling refers to the downward motion of particles driven solely by gravity, affecting both ash and sulfate particles throughout the atmospheric column. In contrast, dry deposition is the surface removal process governed by aerodynamic resistance and surface characteristics (e.g., vegetation, roughness), acting primarily near the surface on particles and gases."

Also, the purple line and gray line are too close in color. I cannot tell which one is which.

There is no grey line in Figure 3. The colors were likely distorted during printing. We recommend viewing the figure on a screen, where the colors are clearly distinguishable.

Whole paper: CTRL and PRTB. What're these abbreviations mean? Your test is not control and perturbed; they are Radiative on and off. So, I would suggest to change to different names that easier for audiences to follow.

Thank you for the suggestion. We have replaced CTRL with RADOFF and PRTB with RADON, corresponding to simulations with radiation feedback turned off and on, respectively.

Line 373: how do you distribute the sulfate and ash amount vertically? uniformly from surface to 1 hPa?

We do not emit sulfate in the long-term experiment. The vertical distribution of ash follows the inverted, height- and time-varying emission scenario derived in Ukhov et al. (2023). We added the following sentence in the text: "Sulfate is not emitted in this experiment."

Line 378: "75 Mt of water vapor" what is this number based on? Citation. Also, you haven't talked about any water progression or radiative impact later on, why do you inject them?

In this experiment, we injected 100 Mt of water vapor, not 75 Mt. This value follows the estimate used in Stenchikov et al. (2021). We added this reference and explained why we included water vapor emission. The updated sentence now reads: "We emit 100 Mt of water vapor, with 75 Mt distributed according to a parabolic profile between 17 and 12 km, and the remaining 25 Mt distributed linearly between 12 and 1 km, following Stenchikov et al. (2021). We inject water vapor to account for its radiative influence on the volcanic plume, as water vapor can modify buoyancy and radiative heating. Although we do not analyze water vapor evolution in this paper, it is included to ensure physically consistent plume thermodynamics."

Line 397: explain aerosol index

We have now added a concise explanation of the Aerosol Index (AI) in the text: "The AI indicates the presence of UV-absorbing aerosols, derived from the contrast between measured and modeled radiances at two ultraviolet wavelengths. Positive AI values correspond to absorbing aerosols such as volcanic ash, whereas near-zero or negative values represent scattering aerosols or clear skies (Krotkov et al., 1999)."

Line 401: "Ukhov et al. (2023) a better agreement between these fields was achieved" Why don't you use what you have before?

In this study, we used the inverted ash and SO2 emission scenarios from Ukhov et al. (2023). However, due to differences in the ash bin size ranges and resulting redistribution of ash mass between optical bins, the agreement with the aerosol index was slightly better in Ukhov et al. (2023).

Figure 6: what's the difference between gravitational settling and dry deposition? I thought they are the same process.

The response was given above.

Line 448: "This number confirms that nearly all of the injected ash (54.17 out of 66.53 Mt) is removed from the atmosphere within 3 months." many papers presented observed ash particles a year after the Pinatubo eruption, seems conflict with your simulation here. Please comment on that.

We thank the reviewer for this important point. The sentence now reads as follows: "The total mass of ash deposited over the domain is 54.18 Mt, confirming that nearly 80% of the injected ash (54.18 out of 66.53 Mt) was removed from the atmosphere within three months. This removal primarily reflects the gravitational settling of coarse ash particles (> 1  $\mu$ m; Stenchikov et al. (2021)), whereas submicron particles remained aloft for a longer period."

```
import numpy as np
import matplotlib.pyplot as plt
**Inputs (your digitized profile)**
z = np.array([0, 1, 2, 3, 4, 5, 6, 7, 8, 9, 10, 11, 12, 13, 14,
        15, 16, 17, 18, 19, 20, 21, 22, 23, 24, 25]) # km
RH = np.array([93.0, 89.0, 87.0, 85.0, 83.0, 81.0, 81.0, 83.0, 85.3,
         87.3, 88.3, 91.0, 94.0, 91.7, 83.0, 73.0, 60.0, 45.0,
         30.0, 18.3, 9.0, 3.3, 1.0, 1.0, 1.0, 1.0]) / 100.0 # fraction
**Convert to meters if needed elsewhere**
z m = z * 1000.0 # meters
**Hygroscopic growth parameters**
**-----**
R_dry = 0.62e-6 # m (dry sulfate radius)
rho acid, rho w = 1800.0, 1000.0 \# kg/m^3
def R_wet_from_kappa(kappa, RH_frac):
  """Köhler-style growth (your Eq. 12)."""
  return R_dry * ((1 + RH_frac*(kappa - 1)) / (1 - RH_frac))**(1/3)
def weight_fraction_from_kappa(kappa, RH_frac):
  """Mass fraction of H2SO4 accounting for densities."""
  R_wet = R_wet_from_kappa(kappa, RH_frac)
  V acid = R dry**3
  V_{water} = R_{wet**3} - R_{dry**3}
  m_acid = rho_acid * V_acid
  m water = rho w * V water
  return m_acid / (m_acid + m_water), R_wet
**Compute for three kappa values**
w1, Rw1 = weight_fraction_from_kappa(1.19, RH) # baseline
w2, Rw2 = weight_fraction_from_kappa(0.5, RH) # lower hygroscopicity
w3, Rw3 = weight_fraction_from_kappa(1.4, RH) # higher hygroscopicity
**-----**
**Plot: 1x3 subplots, shared y**
fig, (ax1, ax2, ax3) = plt.subplots(1, 3, figsize=(15, 7), sharey=True)
**Common Y-axis ticks**
major\_ticks = np.arange(0, 25, 4)
minor_ticks = np.arange(0, 25, 1)
**(a) Relative Humidity**
ax1.plot(RH, z, linewidth=2, label='RH')
ax1.set_xlabel('Relative Humidity (fraction)')
ax1.set_ylabel('Altitude (km)')
ax1.grid(True, linestyle='--', alpha=0.6)
```

```
ax1.set xlim(0, 1)
ax1.set yticks(major ticks)
ax1.set yticks(minor ticks, minor=True)
ax1.tick_params(axis='y', which='minor', length=4, width=0.8)
ax1.set_title('a) Relative Humidity vs Altitude')
**(b) H2SO4 Weight Percent (three kappas)**
ax2.plot(w1*100.0, z, linewidth=2, label='k = 1.19')
ax2.plot(w2*100.0, z, linewidth=2, linestyle='--', label='\kappa = 0.5')
ax2.plot(w3*100.0, z, linewidth=2, linestyle=':', label='\kappa = 1.4')
ax2.set xlabel('H2SO4 Weight Percent (%)')
ax2.grid(True, linestyle='--', alpha=0.6)
ax2.set xlim(0, 100)
ax2.set_yticks(major_ticks)
ax2.set_yticks(minor_ticks, minor=True)
ax2.tick_params(axis='y', which='minor', length=4, width=0.8)
ax2.legend(loc='lower right')
ax2.set title('b) H2SO4—H2O Solution vs Altitude')
**(c) R_wet for three kappas**
ax3.plot(Rw1*1e6, z, linewidth=2, label='\kappa = 1.19')
ax3.plot(Rw2*1e6, z, linewidth=2, linestyle='--', label='\kappa = 0.5')
ax3.plot(Rw3*1e6, z, linewidth=2, linestyle=':', label='k = 1.4')
ax3.set xlabel(r'$R {\mathrm{wet,\,sulf}}$ ($\mu$m)')
ax3.grid(True, linestyle='--', alpha=0.6)
ax3.set_yticks(major_ticks)
ax3.set yticks(minor ticks, minor=True)
ax3.tick_params(axis='y', which='minor', length=4, width=0.8)
ax3.set_title('c) Wet Radius vs Altitude')
ax3.legend(loc='lower right')
**Final layout**
**plt.suptitle('Vertical Profiles: RH, H2SO4 Weight Fraction, and Wet Radius (κ Sensitivity)', fontsize=12)**
plt.tight_layout(rect=[0, 0, 1, 0.96])
plt.show()
```

---

## Author Comment (AC2)

**General comments:**

This paper introduces several key enhancements to the WRF-Chem v4.8 model aimed at improving the simulation of volcanic eruptions, including the implementation of wet and dry deposition for ash and sulfate, SO2 oxidation mechanisms, gravitational settling corrections, and the direct radiative effects of volcanic aerosols. The authors also developed the calculation of ash and aeresol radiation, which has the feedback effects to the meteorology. Using the 1991 Mt. Pinatubo eruption as a case study, the authors evaluate the model's performance through both short-term and long-term experiments, demonstrating clear improvements in mass conservation and a better agreement with satellite observations, particularly when radiative feedback is activated.

Overall, the paper presents a thorough and valuable contribution to the field of volcanic plume modeling. The enhancements address important shorts in WRF-Chem's capabilities. Here recommend minor revisions before publication.

Dear Dr. Mingzhao Liu, we appreciate your positive feedback. Below, we address your comments. Our detailed responses are provided in blue. Changes in the text are in *italic*.

**Main comments:**

1. Fig. 4 and 5 show significant improvements in aerosol and SO2 transport when radiative feedback is included. To further strengthen the model–satellite comparison, the authors should consider applying satellite-specific Averaging Kernels to the model output. This would account for the vertical sensitivity of the satellite retrievals and enable a more rigorous and physically consistent validation.

We appreciate the reviewer's suggestion to apply satellite-specific averaging kernels to the model-satellite comparison. However, the TOMS SO2 retrievals used in this study do not provide per-pixel averaging kernels or vertical weighting functions. The TOMS SO2 product represents a total column amount derived from differential UV backscatter at discrete wavelengths, assuming an effective SO2 layer height (15–25 km for stratospheric plumes such as Pinatubo (Krueger et al., 1995)). Therefore, applying an averaging kernel, as is possible for OMI or TROPOMI products, is not feasible for the TOMS data. Moreover, the inverted emission scenarios (Ukhov et al., 2023) for ash and SO2 were based directly on the TOMS SO2 column loadings and aerosol index (AI) without air-mass-factor corrections.

Krueger, A. J., et al. "Volcanic sulfur dioxide measurements from the total ozone mapping spectrometer instruments." Journal of Geophysical Research: Atmospheres 100.D7 (1995): 14057-14076.

Ukhov, A., et al. "Inverse modeling of the initial stage of the 1991 Pinatubo volcanic cloud accounting for radiative feedback of volcanic ash." Journal of Geophysical Research: Atmospheres 128.12 (2023).

2. Figure 5 illustrates how radiative feedback alters the spatial pattern and magnitude of the SO2 plume. The manuscript would benefit from a more detailed explanation of the underlying physical mechanism. Specifically, how does the absorption of solar radiation by ash influencing SO2 transport and dispersion? A brief discussion linking the radiative heating (e.g., as shown in Fig. 10) to the dynamical response (e.g., enhanced lofting or altered wind patterns) would strengthen the scientific insight of the paper.

We agree, and we have expanded the description of Figure 5 in Section 3.2 by adding the following text: "The absorption of solar radiation by volcanic ash warms the surrounding air within the ash plume, enhancing its buoyancy. This heating also modifies the plume's vertical and horizontal structure. This dynamical response in

the RADON run (Fig. 5b) results in a broader SO2 plume compared to the RADOFF run (Fig. 5c). The altered temperature gradients also modify local wind fields, slightly shifting the transport pathway of SO2 cloud."

3. Some abbreviations are not explicitly defined upon first use, such as LW/SW/PRTB/CTRL.

Corrected. Now, abbreviations are properly defined in the text. We also replaced PRTB by RADON and CTRL by RADOFF, as requested by the 2nd reviewer.

4. In conclusion section, it is claimed that an open-source preprocessor called PrepEmisSources is developed. However, there is no detail introduction to this tool. Please expand it for more details.

In the original manuscript, there is a dedicated section 'Appendix A', which provides a detailed introduction to this tool. More details on how to use the PrepEmisSources utility are presented in: *Ukhov, A. and Hoteit, I.: PrepEmisSources: a framework for preparing volcanic emissions, https://doi.org/10.5281/zenodo.16856541, 2025.* However, we improved the 'navigation' to the 'Appendix A' in several places where the utility is mentioned in the text.

Technical corrections/suggestions:

L. 85: "fixed and error" -> "fixed an error"

**Corrected.**

L. 119: "The updated SO2 concentration(mol mol-1) is calculated": The rate coefficient k is given in units of cm^3 molecule^-1 s^-1. Please verify and ensure unit consistency throughout the calculation

Units analysis shows that the exponential term is dimensionless, which is fine. Thus, this formulation is valid regardless of whether SO2 is expressed in mol/mol or ppmv. Therefore, no mistake here.